# Structures of topoisomerase V in complex with DNA reveal unusual DNA-binding mode and novel relaxation mechanism

Amy Osterman, Alfonso Mondragón*

Department of Molecular Biosciences, Northwestern University, Evanston, United States

**Abstract** Topoisomerase V is a unique topoisomerase that combines DNA repair and topoisomerase activities. The enzyme has an unusual arrangement, with a small topoisomerase domain followed by 12 tandem (HhH)$_2$ domains, which include 3 AP lyase repair domains. The uncommon architecture of this enzyme bears no resemblance to any other known topoisomerase. Here, we present structures of topoisomerase V in complex with DNA. The structures show that the (HhH)$_2$ domains wrap around the DNA and in this manner appear to act as a processivity factor. There is a conformational change in the protein to expose the topoisomerase active site. The DNA bends sharply to enter the active site, which melts the DNA and probably facilitates relaxation. The structures show a DNA-binding mode not observed before and provide information on the way this atypical topoisomerase relaxes DNA. In common with type IB enzymes, topoisomerase V relaxes DNA using a controlled rotation mechanism, but the structures show that topoisomerase V accomplishes this in different manner. Overall, the structures firmly establish that type IC topoisomerases form a distinct type of topoisomerases, with no similarities to other types at the sequence, structural, or mechanistic level. They represent a completely different solution to DNA relaxation.

*For correspondence:
a-mondragon@northwestern.edu

**Competing interest:** The authors declare that no competing interests exist.

## Editor's evaluation

This is a valuable paper with convincing data. The work presents the first structure of Methanopyrus kandleri Topoisomerase V bound to DNA, revealing two important features of the enzyme's mechanism. The first is that the active site toggles between opened, DNA-accessible state to a closed state where the active site cleft is blocked and inaccessible to nucleic acid. The second is a striking array of helix-hairpin-helix motifs that wrap the duplex DNA. The findings will be of interest to researchers working on understanding structure/function relationships of nucleic acid enzymes, particularly in the topoisomerase and DNA repair fields.

## Introduction

The topological state of DNA in cells is regulated by the action of DNA topoisomerases (*Bush et al., 2015*; *Corbett and Berger, 2004*; *Pommier et al., 2016*; *Wang, 2002*). Different cellular process, such as transcription and recombination, can alter the topology of DNA and the action of topoisomerases helps maintain the correct topological state. In order to change the topology of DNA, topoisomerases transiently break either one or two strands of DNA in order to allow movement of strands before the breaks are resealed. In this manner, topoisomerases can relax and supercoil DNA, catenate/decatenate, and knot/unknot DNA molecules, and in some cases RNA as well (*Ahmad et al., 2016*; *Wang et al., 1996*). Due to their involvement in crucial cellular processes, topoisomerases are the target of important chemotherapeutic agents (*Pommier et al., 2010*).

Topoisomerases are found in all three domains of life, but the diversity in structure and sequence suggests that the major subtypes evolved independently (*Forterre et al., 2007*). All topoisomerases have in common the use of phosphotyrosine intermediates for transient cleavage through a transesterification mechanism (*Corbett and Berger, 2004*). Topoisomerases are classified into two types (I and II) based on whether they cleave one or two strands of DNA in a concerted manner. Topoisomerases are subclassified based on similarities at the sequence and structural levels. Type I enzymes, which cleave one DNA strand, are subclassified into three subtypes, IA, IB, and IC. Type IA enzymes are found in bacteria, archaea, and eukarya, employ an enzyme-bridged strand passage mechanism, and share a strand passage structural domain with a very characteristic toroidal shape (*Lima et al., 1994*). Type IB enzymes are also found in all domains of life, employ a swiveling or controlled rotation mechanism, and show clear structural and sequence similarities, despite some of them being much larger than other members of the same subtype (*Corbett and Berger, 2004*). The structure and mechanism of type IA and IB topoisomerases and their cellular role are well studied, and in both subtypes the general steps of the DNA-binding and cleavage/religation mechanism are well understood.

Topoisomerases of the third subtype, IC, have only been found in the archaeal *Methanopyrus* genus (*Forterre, 2006*) and relax DNA using a controlled rotation/swiveling mechanism (*Taneja et al., 2007*). Its only member is topoisomerase V, which was originally described as a type IB enzyme (*Slesarev et al., 1993*), but later classified into its own subtype due to the lack of sequence or structural similarities with other topoisomerases (*Forterre, 2006*; *Taneja et al., 2006*). The exact cellular role of topoisomerase V in the hyperthermophile *Methanopyrus kandleri* is not known. Topoisomerase V works not only at extremely high temperatures (65–122°C), but also in high salt concentrations (*Slesarev et al., 1994*). It is also an unusual protein as it combines DNA repair and topoisomerization activities in the same polypeptide (*Belova et al., 2001*; *Rajan et al., 2016*; *Rajan et al., 2013*), a unique feature of this enzyme. Topoisomerase V is formed by a small, ~30-kDa topoisomerase domain followed by 24 tandem helix–hairpin–helix (HhH) repeats (*Doherty et al., 1996*) arranged as 12 $(HhH)_2$ domains (*Belova et al., 2002*; *Shao and Grishin, 2000*). HhH repeats are associated with DNA lyase repair activity (*Doherty et al., 1996*; *Shao and Grishin, 2000*) and 3 of the 12 $(HhH)_2$ domains have apurinic/apyrimidinic (AP) lyase and deoxyribose-5-phosphate (dRP) lyase activities (*Rajan et al., 2016*; *Rajan et al., 2013*). These 12 $(HhH)_2$ domains also stimulate the topoisomerase activity as mutant topoisomerase V proteins with fewer repeats show decreasing relaxation activity (*Belova et al., 2002*; *Rajan et al., 2010*). $(HhH)_2$ domains are usually found as isolated domains, not in tandem arrangements (*Shao and Grishin, 2000*), which is another unusual characteristic of topoisomerase V. The topoisomerase domain has no sequence or structural similarities to other topoisomerases or any other protein, making it a unique fold (*Forterre, 2006*; *Taneja et al., 2006*). The active site residues have been identified (*Rajan et al., 2014*), and their three dimensional arrangement suggests a different catalytic mechanism from other topoisomerases (*Rajan et al., 2014*). In the structures of the free protein, the topoisomerase domain active site is inaccessible as it is covered by the $(HhH)_2$ domains (*Rajan et al., 2016*; *Rajan et al., 2013*; *Rajan et al., 2010*; *Taneja et al., 2006*).

The way that topoisomerase V interacts with DNA, relaxes it, and recognizes lesions is not known. This stands in contrast to all other topoisomerase types, either I or II, where structures of complexes of the proteins with DNA are known (*Bush et al., 2015*; *Corbett and Berger, 2004*). The absence of structural information on topoisomerase V in complex with DNA has hampered efforts to understand fully not only its mechanism of DNA relaxation, but also the way AP/dRP lyase and topoisomerase activities are coupled and the conformational changes needed to expose the active site and bind DNA. Here, we present structures of topoisomerase V in complex with DNA containing an abasic site. The structures show the way the $(HhH)_2$ domains embrace DNA and change conformation to expose the topoisomerase active site and accept DNA into it. Exposure of the active site involves breaking of a long helix linking the topoisomerase and $(HhH)_2$ domains. The novel manner employed by topoisomerase V to bind DNA suggests that the $(HhH)_2$ domains serve as a processivity factor in addition to containing the repair active sites. Based on the structural findings, important regions were identified and site-directed mutagenesis studies probed the role of different residues in DNA relaxation. Despite type IB and IC enzymes both using a controlled rotation DNA relaxation mechanism (*Koster et al., 2005*; *Taneja et al., 2007*), the structures show that there are significant and substantial differences; topoisomerase V bends DNA sharply at the active site resulting in melting of the DNA, which may facilitate strand rotation. The structures establish that type IC enzymes share no features with other

**Table 1.** Crystallization conditions.

| Dataset | Topo V:DNA concentration | Crystal growth conditions | Native or derivative with soak length | Cryoprotectant |
| --- | --- | --- | --- | --- |
| Topo-97(ΔRS2) with 38 bp asymmetric DNA | 32:40 µM | 30°C, 1:1 µl (reaction:well solution), 9–10% PEG 600, 50 mM sodium succinate pH 5.5, 200 mM potassium chloride, 10 mM magnesium chloride, 1 mM spermine | Native Derivative: 1 mM phosphotungstic acid, 2 min | 25–30% PEG 400 |
| Topo-97(ΔRS2) with 38 bp asymmetric DNA and no abasic site | 48:60 µM | 30°C, 2:1 µl (reaction:well solution), 10% PEG 400, 50 mM MES pH 5.6, 200 mM potassium chloride, 10 mM magnesium chloride, 1 mM spermine | Native | 35% PEG 400 |
| Topo-97(ΔRS2) with 38, 39 or 40 bp symmetric DNA | 48:60 µM | 30°C, 2:1 µl (40 bp) or 2:2 µl (38, 39 bp) (reaction:well solution), 2% PEG 8K, 24 mM sodium acetate pH 5.1, 26 mM sodium acetate pH 5.6, 12.5 µM phosphotungstic acid | Native: 38 bp (high resolution), 40 bp, 39 bp. Native*: 38 bp with 2.5 mM KAuCl₄ (low resolution native) Derivative: 38 bp with 350–400 µM undecagold, 6–8 min | 35% PEG 400 |

*Crystal was soaked in $KAuCl_4$, but derivative was not detected and was treated as native.

type of topoisomerases at any level. They form a distinct subtype employing a different mechanism of DNA cleavage/religation and relaxation. The structures not only bring our understanding of type IC enzymes to a similar level as for all other topoisomerases, but also demonstrate a different mechanism of DNA relaxation that has not been observed before. Topoisomerase V, due to its unusual characteristics, continues to expand our understanding of mechanisms of DNA topological manipulations and exemplifies a solution to the way proteins relax DNA that is different from the ones employed by other topoisomerases.

## Results

### Crystallization of complexes of topoisomerase V with DNA

Topoisomerase V is a nonsequence-specific DNA-binding protein that has AP lyase activity and interacts with DNA abasic sites (*Rajan et al., 2016*; *Rajan et al., 2013*). For this reason, co-crystallization trials were done using DNA fragments with abasic sites with the goal to target topoisomerase V to this position and create a more homogeneous complex. In addition, a 97-kDa fragment of topoisomerase V mutated to have only a single AP lyase site (hereafter Topo-97(ΔRS2), see Materials and methods) (*Rajan et al., 2016*) was used in all crystallization trials. After trying several DNA sequences, lengths, and overhangs, the optimal crystals grew with a 38-base pair (bp) DNA oligonucleotide with a single abasic site and two complementary overhanging base pairs at the ends of the DNA (Materials and methods and *Table 1*). This structure is in a closed conformation.

The 38-bp DNA oligonucleotide was redesigned by making it symmetric from the center, with two abasic sites, and still containing overhangs (total length per strand 40 nucleotides) (Materials and methods and *Table 1*) and led to a different crystal form. The structure of the symmetric oligonucleotide complex shows that each DNA fragment binds two protein molecules with the topoisomerase active site accessible and occupied by DNA. The two monomers are related by a twofold axis, but this axis does not coincide with the DNA sequence twofold axis creating an asymmetry in the DNA structure. A second symmetric DNA molecule (39 bp long) was designed with the twofold axis passing through a central base and produced a structure with a symmetric DNA molecule and with the protein and DNA twofold axes coinciding. A third and slightly longer DNA molecule (40 bp) based on the 38-bp symmetric oligonucleotide resulted in slightly better diffraction.

### Overall structure of topoisomerase V with asymmetric DNA

Topo-97(ΔRS2) consists of the topoisomerase domain at the N-terminal end and 10 (HhH)₂ domains arranged in tandem (*Figure 1A*). The crystals of the complex formed by Topo-97(ΔRS2) with a 38-bp asymmetric DNA oligonucleotide containing one abasic site diffract to a 3.24 Å in the best direction

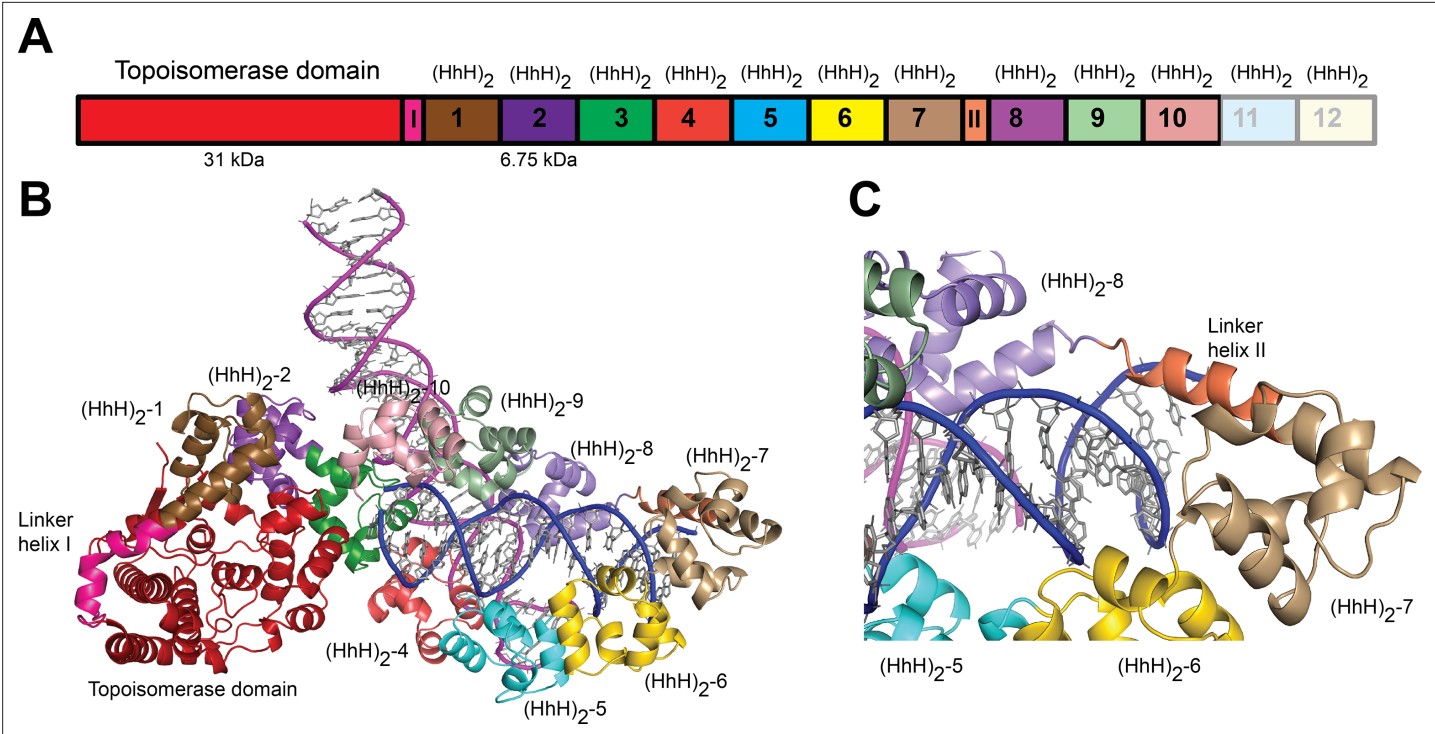

**Figure 1.** Structure of topoisomerase V in complex with asymmetric DNA. (**A**) Schematic diagram of the domain organization of topoisomerase V. The protein contains a small, 31-kDa topoisomerase domain followed by twelve (HhH)$_2$ domains each formed by two (HhH) repeats. There are linker helices between the topoisomerase and the first (HhH)$_2$ domain (LI) as well as between repeats 7 and 8 (LII). Repeats 11 and 12 are not part of the structure. (**B**) Cartoon showing the structure of topoisomerase V in complex with asymmetric DNA. Each protein monomer binds two different half DNA molecules. The two different DNA molecules are shown colored in pink and blue. The topoisomerase domain remains blocked by the (HhH)$_2$ domains preventing access to the active site. (**C**) Closeup view of the region around repeats 7 and 8, which are connected by Linker helix II. The linker helix serves to connect two sets of domains and sits above the major groove of the DNA. In this structure, one DNA is surrounded by the (HhH)$_2$ domains while the other sits between (HhH)$_2$ domains. The domains are colored using the same scheme as in the (**A**) diagram.

The online version of this article includes the following figure supplement(s) for figure 1:

**Figure supplement 1.** Dimers in the crystal of the topoisomerase V with asymmetric DNA complex.

but only to ~3.9 Å in the other directions (**Table 2**). This anisotropy resulted in an electron density map of uneven quality and although diffraction extends to higher resolution, the map mostly corresponds to a medium resolution structure. Nevertheless, the placement of the protein molecules was unambiguous and a previously unseen region, repeat 7 and Linker helix II, linking repeats 7 and 8, could be seen clearly (**Figure 1**). The region was built with the aid of the structure from the symmetric complexes, which diffract to higher resolution. The DNA molecules were easily recognized and built. The asymmetric unit of the crystal contains two complexes, each formed by a Topo-97(ΔRS2) protein molecule and two different DNA molecules. This arrangement is unusual, with each protein monomer bound to only half of each DNA molecule. The center of two of the half-bound DNA molecules sits on crystallographic twofold axes, recreating the full-length DNA (**Figure 1—figure supplement 1**). The second DNA molecule is bound by the two protein monomers. The two complexes in the asymmetric unit are very similar, but not identical (root mean square deviation [rmsd] between Cα atoms ~1.0 Å).

Each complex shows the protein in the closed conformation with the (HhH)$_2$ domains surrounding one DNA molecule. The second DNA molecule interacts with a subset of (HhH)$_2$ domains in an unanticipated manner (**Figure 1B**). The two DNA molecules bind the protein almost perpendicularly, at a ~120° angle (**Figure 1B**). In previous structures of fragments of topoisomerase V (**Rajan et al., 2016**; **Rajan et al., 2013**), the sixth and seventh (HhH)$_2$ domains and the linker joining it to the eighth one were partially or fully disordered. In the complex structure this region is ordered in one of the monomers and shows that the seventh repeat forms an (HhH)$_2$ domain followed by a long helix (Linker helix II) that connects it to the eighth (HhH)$_2$ domain. This long helix sits in the major groove of one DNA molecule, making the (HhH)$_2$ domains wrap around the DNA (**Figure 1C**). Aside from the ordering of

**Table 2.** Data collection and phasing statistics for asymmetric complex.

| Data collection | Refinement | SIRAS phasing | |
|---|---|---|---|
| | High-resolution native | Native | Phosphotungstic acid |
| Detector type/source | RayonixCCD/APS | RayonixCCD/APS | RayonixCCD/APS |
| Wavelength (Å) | 0.97856 | 0.97856 | 0.97872 |
| Resolution range* (Å) | 39.3–3.24 (3.1–3.24) | 29.97–6.0 (6.71–6.0) | 39.7–6.0 (6.71–6.0) |
| Space group | $P4_12_12$ | $P4_12_12$ | $P4_12_12$ |
| $a = b$, $c$ (Å) | 193.75, 245.9 | 196.17, 246.21 | 198.67, 245.98 |
| Measured reflections* | 428,733 (24,791) | 86,099 (25,413) | 70,747 (20,431) |
| Unique reflections* | 52,587 (2629) | 12,420 (3489) | 12,721 (3557) |
| Spherical completeness* (%) | 70.4 (13.2) | 99.1 (100.0) | 99.2 (99.9) |
| Ellipsoidal completeness* (%) | 96.3 (76.9) | – | – |
| Anomalous completeness* | – | 99.5 (100.0) | 97.8 (99.0) |
| Mean [$I/\sigma(I)$]* | 13.5 (1.7) | 20.4 (4.5) | 17.0 (2.9) |
| Multiplicity* | 8.2 (9.4) | 6.9 (7.3) | 5.6 (5.7) |
| Anomalous multiplicity* | – | 3.8 (3.8) | 3.0 (3.0) |
| $R_{measure}$* | 0.1 (1.567) | 0.067 (0.543) | 0.072 (0.791) |
| $R_{merge}$* | 0.093 (1.48) | 0.057 (0.468) | 0.059 (0.654) |
| CC (1/2) | 0.999 (0.58) | 0.998 (0.812) | 0.998 (0.812) |
| MFID[†] | – | – | 0.336 |
| | | | |
| Phasing | | | |
| Number of sites | – | | 10 |
| Phasing power | | | |
| Dispersive (centric/acentric) | – | | 1345/1.612 |
| Anomalous (acentric) | – | | 1.612 |
| FOM (centric/acentric) | | 0.3486/0.4463 | |

*All numbers in parenthesis are for highest resolution shell.
[†]Mean fractional isomorphous difference against Native.

this region and the swiveling of the last three (HhH)$_2$ domains, no other major changes are seen in the protein with respect to the structure of the same fragment in the absence of DNA (*Rajan et al., 2016*). Thus, in the closed conformation the main change in the protein structure is the movement of (HhH)$_2$ repeats 8–10 to enclose the DNA and the ordering of the seventh domain and the linker region.

The protein binds the two DNA molecules through the (HhH)$_2$ domains. One of the DNA molecules is almost surrounded by the (HhH)$_2$ domains with one end abutted against the topoisomerase domain (*Figure 1B* and *Figure 1—figure supplement 1*). As the enzyme is in the closed conformation, the DNA cannot interact with the topoisomerase domain or enter the active site. The second DNA molecule sits in a positively charged groove formed by the (HhH)$_2$ domains. The two DNA molecules come close together at this point but do not interact directly. The DNA molecules are in the B-conformation with one of them bent, but mostly having canonical DNA parameters. Interestingly, the same DNA sequence without the abasic site also crystallized under the same conditions, suggesting that the presence of the DNA abasic site had no effect on the binding of the protein or the conformation of the DNA. Not surprisingly, the DNA abasic site was not apparent in the structure and does not appear to cause any deviations from canonical B-DNA.

## Overall structure of topoisomerase V with symmetric DNA

The use of a symmetric DNA with two abasic sites resulted in a new crystal form with one DNA molecule bound by two Topo-97(ΔRS2) monomers (*Figure 2—figure supplement 1*). Different lengths of DNA were tried to improve the crystal quality, but all of them suffered from anisotropic diffraction. The best diffraction was from crystals with a 40-bp DNA oligonucleotide with two base overhangs at each end (Materials and methods). These crystals diffract to 2.92 Å in the best direction but only to ~3.5 Å in the other directions (*Table 3*). Crystals with 39 or 40 bp DNA and including two base overhangs were also anisotropic, but served to provide information on the path of the DNA.

In the structure, one DNA molecule is surrounded by two protein monomers through the (HhH)$_2$ domains (*Figure 2A, C*). The two proteins in the asymmetric unit are very similar in conformation and are related by an almost perfect noncrystallographic twofold axis (rmsd between Cα ~ 1.0 Å). For this reason, only one monomer is described hereafter. The (HhH)$_2$ domains can be divided into two subsets, (HhH)$_2$ domains 1–7 and 8–10. Each (HhH)$_2$ domains subset has the same conformation as observed in the DNA-free protein (*Rajan et al., 2016*), showing that they move as rigid groups. Similar to the asymmetric DNA complex, the (HhH)$_2$ domains change direction after the seventh domain and the helix linking the seventh and eighth domains is well ordered. The two subsets form a loop-like structure where the turn is formed by the helix connecting domains 7 and 8. The topoisomerase domain is shifted with respect to the (HhH)$_2$ domains and is accessible to the DNA. The topoisomerase domain moves as a rigid body with the only changes observed confined to the helix linking the topoisomerase domain and the first (HhH)$_2$ domain (Linker helix I) (*Figure 2B*). This helix, which in the DNA-free structure is an extension of the first helix of the (HhH)$_2$ domain, is broken and in this way separates the topoisomerase domain from the (HhH)$_2$ domains by rotating the topoisomerase domain by close to 180°. (*Figure 2—figure supplements 2 and 3*). The surface area at the interface between the topoisomerase domain and the (HhH)$_2$ domains is reduced considerably once the domains separate (*Figure 2—figure supplement 3*), suggesting that the closed form is more stable in the absence of DNA/protein interactions. Interestingly, in both conformations the residues at the interface are mostly hydrophilic amino acids, which is consistent with amino acids that can be either buried or exposed in the two conformations. The change in this helix is the only major change observed when the DNA-free and DNA-bound structures are compared. The movement of the topoisomerase domain exposes the active site (*Figure 2*), which in all other structures was inaccessible, and allows the end of the DNA molecule to enter the topoisomerase domain and interacts with residues near the active site region (*Figure 2D*).

The structure with a 38 bp oligonucleotide revealed that the noncrystallographic twofold axis passes through a base pair of the DNA molecule, making the 38 and 40 bp molecules sit asymmetrically in the complex; one half comprises 18 or 19 bp and the other half 19 or 20 bp for the 38 and 40 bp DNA molecules, respectively. This asymmetry translates in an asymmetry in the path of the DNA, even though the protein monomers are identical. To regularize the complex, a DNA molecule with an odd number of base pairs (39 bp) was used. In this case, the twofold axis passes through the central base pair, the two halves of the DNA molecule are identical, and the path of the DNA is symmetric. The more symmetric structure did not show any differences in the proteins.

The structures with symmetric DNA show a dimer in the asymmetric unit, but from previous small angle X-ray scattering (SAXS) experiments there is no evidence that the free protein dimerizes in solution (*Rajan et al., 2016*). For this reason, the existence of a dimer in solution, not in the crystals, was investigated further. Dynamic light scattering (DLS) measurements with the full-length protein showed residual protein aggregation, precluding good size estimates. DLS experiments (see Materials and methods) with 40-bp symmetric DNA and the Topo-97(ΔRS2) protein were consistent with a monomeric structure, that is, one protein bound per DNA molecule. The DLS data for the Topo-97(ΔRS2) complex with 40-bp symmetric DNA shows an average diameter of 104 Å and an estimated molecular weight of 137 kDa, consistent with a 97 kDa protein plus ~26.5 kDa for the DNA. For comparison, calculation of the expected hydrated diameter of the complex using HullRad (*Fleming and Fleming, 2018*) indicates ~104 and ~120 Å for the monomer and dimer, respectively. The free Topo-97(ΔRS2) protein has a calculated hydrated diameter of 95 Å, consistent with the 94 and 92 Å diameter obtained from DLS and SAXS (*Rajan et al., 2016*) measurements, respectively. The complex of Topo-97(ΔRS2) complex with 38-bp asymmetric DNA shows a diameter of 99 Å, but a molecular weight of ~161 kDa, which is consistent with one protein and two DNA molecules

**Table 3.** Data collection and phasing statistics for symmetric complexes.

**Data collection**

| Dataset | Refinement datasets | | | Phasing data | |
|---|---|---|---|---|---|
| | 40-bp symmetric DNA | 39-bp symmetric DNA | 38-bp symmetric DNA | 38-bp symmetric DNA | Undecagold |
| | Native | Native | Native | Native | |
| Detector type/source | RayonixCCD/APS | RayonixCCD/APS | RayonixCCD/APS | Eiger 9 M/APS | Eiger 9 M/APS |
| Wavelength (Å) | 0.97872 | 0.978720 | 0.97872 | 1.0331 | 1.0357 |
| Resolution range* (Å) | 108.8–2.92 (3.24–2.92) | 86.9–3.17 (3.36–3.17) | 59.1–3.52 (3.73–3.52) | 49.7–3.38 (3.67–3.38) | 40.0–3.45 (3.64–3.45) |
| Space group | $P4_32_12$ | $P4_32_12$ | $P4_32_12$ | $P4_32_12$ | $P4_32_12$ |
| a = b, c (Å) | 120.94, 497.57 | 121.56, 497.19 | 121.67, 498.82 | 121.17, 488.14 | 120.97, 497.69 |
| Number of crystals | 1 | 1 | 7 | 1 | 5 |
| Measured reflections* | 782,879 (51,701) | 470,215 (26,431) | 398,382 (13,495) | 928,796 (56,880) | 5,726,189 (273,394) |
| Unique reflections* | 56,144 (2768) | 53,311 (2667) | 41,745 (2087) | 40,840 (2042) | 44,699 (2236) |
| Spherical completeness* (%) | 69.0 (12.9) | 82.4 (25.7) | 87.5 (28.2) | 78.0 (17.8) | 89.2 (29.8) |
| Ellipsoidal completeness* (%) | 94.7 (69.6) | 92.0 (68.6) | 94.7 (54.4) | 94.5 (68.5) | 95.4 (51.9) |
| Anomalous completeness* | – | – | – | 94.3 (67.0) | 95.1 (50.5) |
| Mean $I/\sigma(I)$* | 21.0 (1.7) | 18.1 (1.5) | 12.0 (1.7) | 8.0 (2.1) | 12.3 (1.8) |
| Multiplicity* | 13.9 (18.7) | 8.8 (9.9) | 9.5 (6.5) | 22.7 (27.9) | 128.1 (122.3) |
| Anomalous multiplicity* | – | – | – | 12.3 (15.2) | 69.4 (65.8) |
| $R_{measure}$* | 0.084 (1.65) | 0.084 (1.41) | 0.128 (1.289) | 0.278 (1.84) | 0.752 (17.66) |
| $R_{merge}$* | 0.081 (1.605) | 0.079 (1.33) | 0.121 (1.183) | 0.272 (1.80) | 0.749 (17.59) |
| CC (1/2) | 1.0 (0.744) | 1.0 (0.646) | 0.997 (0.633) | 0.994 (0.741) | 0.999 (0.749) |
| MFID† | – | – | | – | 0.36 |
| **Phasing** | | | | | |
| Number of sites | | | | | 1 |
| Phasing power | | | | | |
| Dispersive (centric/acentric) | | | | | 3.122/2.755 |
| Anomalous (acentric) | | | | | 0.758 |
| FOM (centric/acentric) | | | | 0.133/0.111 | |

*All numbers in parenthesis are for highest resolution shell.

†Mean fractional isomorphous difference against Native.

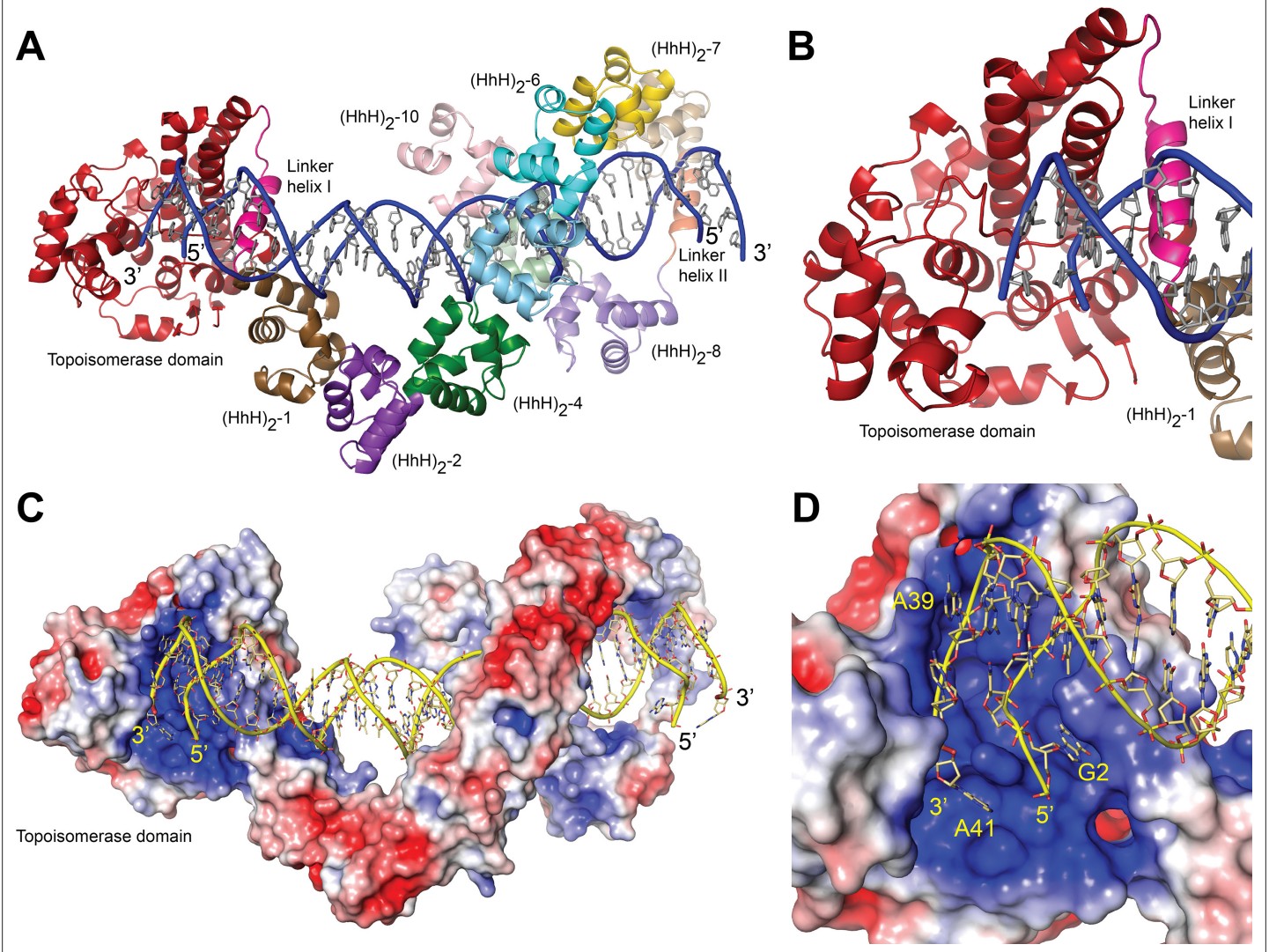

**Figure 2.** Structure of topoisomerase V in complex with symmetric DNA. (**A**) Cartoon showing the structure of topoisomerase V in complex with symmetric 40 bp DNA. Each DNA molecule is bound by two protein molecules in a symmetric manner; only one protein molecule is shown. In the structure, the topoisomerase domain moves away from the (HhH)$_2$ domains allowing access to the active site. The (HhH)$_2$ domains wrap around the DNA. (**B**) Closeup view of the topoisomerase domain (red). Linker helix I (pink) changes conformation to allow movement of the domains and expose the active site. (**C**) Electrostatic surface of topoisomerase V in the bound conformation. The interior of the cavity formed by the (HhH)$_2$ domains is slightly positively charged, forming a region where the DNA can bind. The end of the DNA molecule enters the topoisomerase active site. (**D**) Closeup view of the topoisomerase domain active site. The DNA enters a highly positively charged groove where the active site residues are located. To enter the active site, the DNA bends and the base pairing between bases is broken. The protein domains are colored as in *Figure 1*. The surface is colored with a blue to red gradient from +5 to −5 $K_bT/e_c$.

The online version of this article includes the following figure supplement(s) for figure 2:

**Figure supplement 1.** Dimer in the crystal of topoisomerase V with symmetric 39 bp DNA.

**Figure supplement 2.** A conformational change in Linker helix I exposes the topoisomerase active site.

**Figure supplement 3.** Breaking of the linker helix separates the topoisomerase V domains.

**Figure supplement 4.** Model of the full-length topoisomerase V based on the structure of the 97 kDa fragment in complex with symmetric 39 bp DNA.

(~140 kDa). For comparison, a protein dimer with one DNA molecule bound would have an estimated molecular weight of 217 kDa, very different from any of the values observed. Furthermore, modeling of the missing (HhH)$_2$ domains in the protein using AlphaFold2 (*Jumper et al., 2021*) show that the full-length protein cannot bind as a dimer in the same manner as the 97 kDa fragment in the crystals (*Figure 2—figure supplement 4*). The modeling shows that the last two (HhH)$_2$ domains

would collide with each other in a dimer. The modeling suggests that the full-length protein binds DNA as a monomer whereas the DLS data suggest that in solution the preferred binding mode is also a monomer. It is likely that the observed crystallographic dimer is a result of the crystallization conditions.

## Structure of the DNA in the complexes

In the asymmetric complex each protein monomer interacts with two different DNA molecules. One of them is slightly bent and is shared by two proteins, while the second one is straighter. In either case, it was not possible to observe the abasic site. Unlike the asymmetric complex, the DNA in the symmetric complex shows clearly the presence of the abasic site. The structure shows that the abasic site is accommodated by unstacking the complementary base, which moves to lie in the major groove of the DNA (*Figure 3*). The sugar of the abasic site remains in the expected position. The result of the rearrangements is that the base adjacent to the abasic site moves to occupy the vacant space left by the abasic site. These movements creates a kink in the DNA leading to an opening of the major groove and a narrowing of the major groove just before the abasic site.

The bending around the abasic site allows the DNA to approach the active site (*Figure 2*). In addition, in order to interact with the protein the DNA is very sharply bent where it enters the active site. This sharp bend makes the two DNA strands melt and the last two complementary nucleotides are not base paired (T1:A39,G2:C38) (*Figure 3*). Instead, C38 is unstacked and enters the minor groove, whereas A39 stacks on top of G37 (*Figure 3*). The melting allows the last four nucleotides on the 3′ end of the DNA to enter the protein and do not interact with the other DNA strand (*Figure 3*). The 5′ end of the DNA has two unpaired nucleotides, the first one mostly disordered as it has moved away from the protein whereas the second nucleotide, G2, enters a pocket in the protein, stacked against the side chains of Tyr289 and Arg109.

## DNA–protein interactions

The (HhH)$_2$ domains surround most of the DNA coming in close contact with the phosphate backbone at many points. They form a loop with a positively charged interior that accommodated the DNA, but appears to have few close contacts (*Figure 2*). There are no contacts of the (HhH)$_2$ domains with the bases, only with the backbone, which is not unusual for a nonsequence-specific DNA-binding protein. Repeats 3–6 and 9 make contacts with the phosphate backbone while the rest of the repeats only surround it, but do not come close to it. Interestingly, even though many of the repeats do not contact the DNA directly, they have positively charged residues facing the DNA, creating an overall positively charged environment around the DNA (*Figure 2*).

The abasic sites are not in direct contact with the protein even though the abasic sites were introduced as a possible target for the single intact repair domain. Repeat 6 contains an AP/dRP lyase active site that includes lysines 566, 570, and 571; mutations of any of these three residues are deleterious for activity (*Rajan et al., 2013*). In the complex structure, lysines 570 and 571 face the DNA phosphate backbone, but are not close enough to contact it. Lysine 566 faces away from the DNA and cannot contact the DNA, suggesting that either lysine 570 or 571 are the likely nucleophile. The abasic site is in close contact to repeat 9, which has not been implicated in AP/dRP lyase activity (*Rajan et al., 2013*). It is likely that the protein scans the DNA for lesions by sliding along the DNA. As the protein slides, different DNA regions would interact with the different repair domains facilitating recognition and processing of lesions.

The active site is exposed in the structure and reveals a highly positively charge region where the DNA enters. As mentioned above, the DNA in this region melts due to a sharp bend. This groove is narrow but expands on the side that faces the solvent and where the DNA exits. Binding of DNA in this region is plastic. Oligonucleotides of different lengths can be accommodated by following a slightly different path before entering the active site (*Figure 3—figure supplement 1*). The 5′ end of one strand always follows the same path whereas the 3′ end does not. The last nucleotide on the 3′ end always ends up near the active site, but the unpaired nucleotides before the last one can follow slightly different paths. The contacts between the protein and the DNA are mostly with the phosphate backbone, with the bases facing the solvent, although one nucleobase at the 5′ end is tightly wedged between two protein side chains.

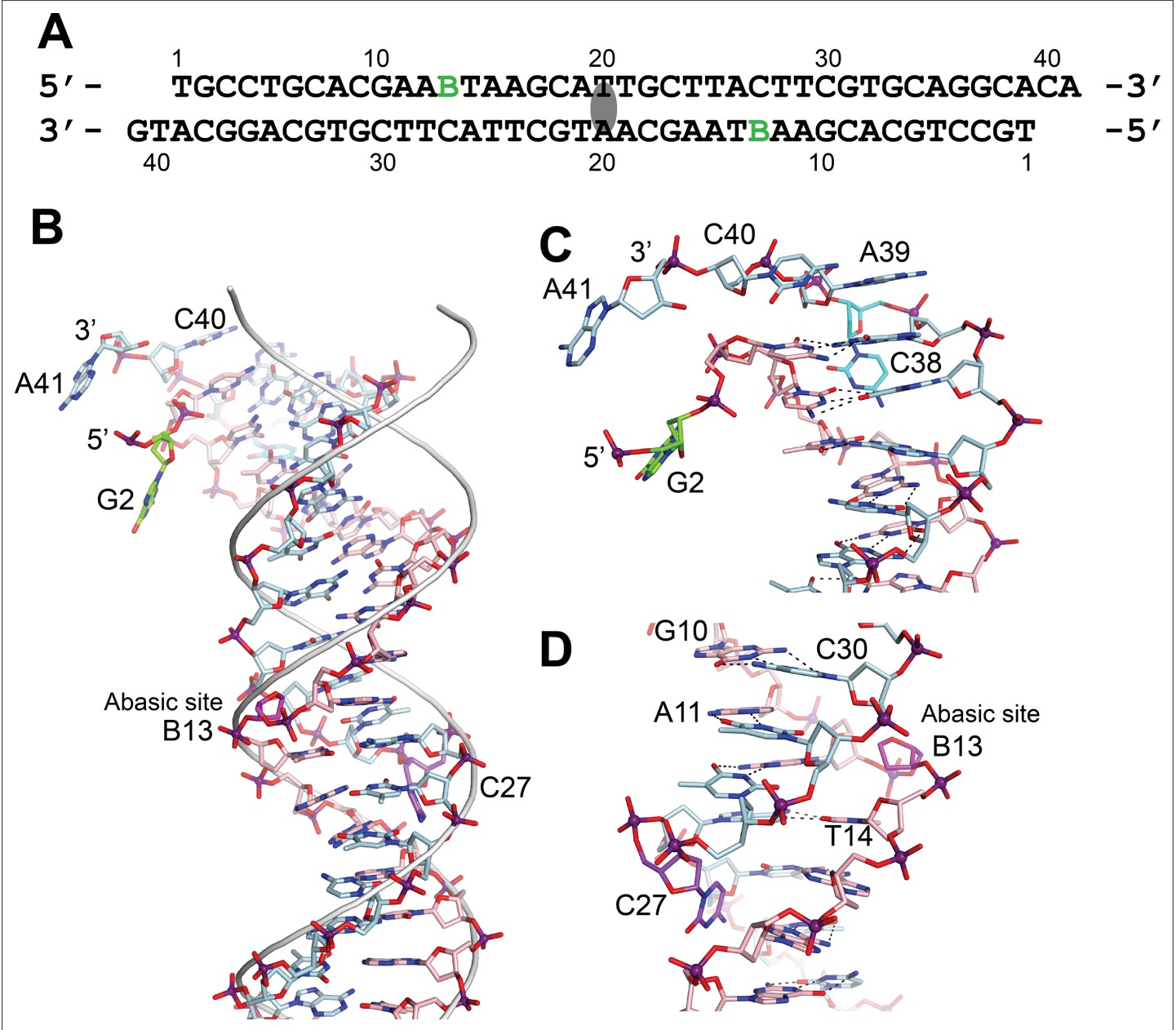

**Figure 3.** DNA in the structure of topoisomerase V with 39 bp DNA. (**A**) Sequence of the 39 bp DNA in the structure. The DNA is symmetric, with the twofold axis passing through the central base pair. The abasic sites are shown with a green B. (**B**) Stick diagram of one half of the 39 bp DNA in the complex. The DNA molecule is bent due to the presence of the abasic sites. For comparison, the phosphate backbone path of a B DNA molecule is shown as a grey tube. Note that the bending occurs around position B13, the abasic site. For clarity, the carbon atoms in one strand are colored in light blue and on the complementary strand in pink. (**C**) Closeup view of the end of the molecule. The DNA is bent where it enters the active site and the bending causes the base paring to break. The second nucleotide, G2 (green), is not base paired to its corresponding base pair, C38 (cyan). The latter unstacks from the helix to allow the next nucleotide A39, to interact with the protein. The first nucleotide, T1, is disordered in the structure. (**D**) Closeup view of the region around the abasic site. The abasic site (B13, magenta) causes the DNA to bend. The corresponding base pair, C27 (purple), completely unstacks from the helix and the base enters the minor groove. The unstacking of C27 plus the movement of the sugar in the abasic site allows the stacking of bases to continue without any gaps, despite the presence of the abasic site.

The online version of this article includes the following figure supplement(s) for figure 3:

**Figure supplement 1.** The protein can accommodate different length DNA entering the active site.

**Table 4.** Refinement statistics for all complexes.

| Refinement | 40-bp symmetric DNA | 39-bp symmetric DNA | 38-bp symmetric DNA | 38-bp asymmetric DNA (no abasic site) |
|---|---|---|---|---|
| Resolution (Å) | 58.76–2.92 | 59.04–3.17 | 59.10–3.52 | 39.28–3.24 |
| Number of reflections working/test* | 56,123/2845 | 53,291/2690 | 41,720/2217 | 52,466/2599 |
| $R$ (working set; %)[†] | 25.71 (43.55) | 21.84 (34.04) | 22.29 (33.41) | 23.35 (39.48) |
| $R_{free}$ (test set; %)[‡] | 28.77 (26.08) | 25.11 (69.52) | 27.00 (38.76) | 26.38 (25.78) |
| Structure quality | | | | |
| Protein atoms | 13,614 | 13,604 | 13,604 | 13,014 |
| DNA atoms | 1606 | 1619 | 1561 | 2868 |
| Other atoms | 32 | 4 | 5 | 44 |
| RMS deviations in bond lengths (Å) | 0.002 | 0.002 | 0.002 | 0.004 |
| RMS deviations in bond angles (°) | 0.489 | 0.442 | 0.421 | 0.540 |
| Average $B$ factor (Å$^2$) (DNA) | 110.87 | 117.99 | 138.26 | 191.74 |
| Average $B$ factor (Å$^2$) (protein) | 97.13 | 104.43 | 129.73 | 135.94 |
| Ramachandran plot[§] | | | | |
| Favored regions (%) | 97.52 | 95.81 | 95.99 | 97.22 |
| Outliers (%) | 0.24 | 0.41 | 0.41 | 0 |
| Distribution $Z$-score | −1.99 | −1.90 | −1.27 | −1.51 |
| Clashscore | 5.93 | 4.76 | 4.94 | 4.86 |
| Molprobity score | 1.42 | 1.53 | 1.53 | 1.4 |

*Numbers in parenthesis correspond to highest resolution shell.

[†]$R_{work} = \Sigma\,||F_o| - |F_c||/\,\Sigma|F_o|$, where $|F_o|$ is the observed structure factor amplitude and $|F_c|$ the calculated structure factor amplitude.

[‡]$R_{free} = R_{factor}$ based on 5% of the data excluded from refinement.

[§]As reported by Molprobity (**Davis et al., 2004**).

## Site-directed mutagenesis studies support the structural observations

In order to assess the role of different residues in guiding DNA to the active site and also the importance of the conformational change in the linker region, several amino acids were mutated and the relaxation activity of the mutants was compared with that of wild-type Topo-97. The comparison was done based on DNA relaxation activity assays and was qualitative. All experiments were done in triplicates as described in Materials and methods. Three areas were probed: (1) around the positively charged groove leading to the active site and where the bending and melting of DNA occurs, (2) the area surrounding the 5′ end of the noncleaved DNA strand and adjacent to the active site, and (3) the region in the linker helix that breaks to expose the topoisomerase domain. *Table 4*; *Table 5* shows the mutants tested and the results. *Figure 4* shows the location of the mutants in the structure and the results. Typical data are shown in *Figure 4—figure supplement 1*.

Five positively charged residues in the region leading to the active site were mutated to alanine. Two of them showed noticeably reduced activity whereas three of them show wild-type levels of activity. Interestingly, a double mutant with two positively charged residues mutated to alanine showed no activity, suggesting that while the enzyme can accommodate small charge changes, more extensive changes abolish the activity and confirming the importance of the positively charged region. Another mutant was designed to test the structure of a loop that abuts against the cleaved DNA strand; Ala132 faces the interior of the protein and replacing it with a bulkier isoleucine (Ala132Ile) reduced very significantly the activity, suggesting that the structure of this region is important for DNA binding. The results confirm the importance of the overall charge of the region, but more importantly they confirm that the structure of the protein in this area is crucial for activity.

**Table 5.** Mutants to probe different regions in the protein[*].

| Mutant | Region | Activity[†] | Source[‡] |
|---|---|---|---|
| Arg37Ala | Positively charged region leading to active site | Reduced | Synthesized |
| Lys47Ala | Positively charged region leading to active site | Wild type | Synthesized |
| His56Ala | Positively charged region leading to active site | Wild type | Synthesized |
| Ala132Ile | Positively charged region leading to active site | Minimal | Synthesized |
| Lys134Ala | Positively charged region leading to active site | Reduced | Synthesized |
| Arg135Ala | Positively charged region leading to active site | Wild type | Synthesized |
| Lys134Ala, Arg135Ala | Positively charged region leading to active site | Minimal | Synthesized |
| Lys134Glu, Arg135Glu | Positively charged region leading to active site | None | Site-directed mutagenesis |
| Arg83Ala | Interacting with noncleaved DNA strand | Minimal | Synthesized |
| Arg108Ala | Interacting with noncleaved DNA strand | None | Synthesized |
| Arg109Ala | Interacting with noncleaved DNA strand | Reduced | Site-directed mutagenesis |
| Arg108Ala, Arg109Ala | Interacting with noncleaved DNA strand | None | Synthesized |
| Arg288Ala | Linker helix | Wild type | Synthesized |
| Tyr289Ala | Linker helix | Wild type | Synthesized |
| Leu290Pro | Linker helix | Minimal | Site-directed mutagenesis |
| Arg293Ala | Linker helix | Wild type | Synthesized |
| Arg288Ala, Arg293Ala | Linker helix | Reduced | Site-directed mutagenesis |
| Arg288Glu, Arg293Glu | Linker helix | None | Site-directed mutagenesis |
| Arg288Glu, Leu290Pro, Arg293Glu | Linker helix | None | Site-directed mutagenesis |

[*]Activity: Level of DNA relaxation activity was assessed qualitatively by comparing against the activity levels of the Topo-97 wild-type enzyme.

[†]Source: Mutations were introduced either by site-directed mutagenesis or by total synthesis of the mutant. All mutations were done in the Topo-97 wild-type backbone.

[‡]Table entries colored according to areas delineated in **Figure 4**.

Residues near the active site, but mostly in contact with the noncleaved strand, had a more marked effect on DNA relaxation activity. Arg108 lies between the two DNA strands and mutating it to alanine completely abolishes activity, suggesting that Arg108 is an essential residue. Both Arg83 and Arg109 face the noncleaved strand and mutating them separately to alanine resulted in reduced relaxation activity with Arg83Ala showing a much lower level of activity. Combining Arg108Ala and Arg109Ala completely abolishes activity, which is not surprising as R108 is essential. These results underlie the importance of the interactions with the intact strand and uncover a residue, Arg108, as essential for activity.

Finally, residues in the linker helix were mutated to study their importance in the relaxation reaction. Mutating Arg288, Tyr289, and Arg293 to alanine had no effect on activity. Both Arg288 and Tyr289 face the major groove of DNA and Tyr289 and Arg109 form a pocket where a nucleobase sits. Arg293 contacts the DNA phosphate backbone directly, but far away from the active site. Mutating simultaneously both Ar288 and Arg293 to alanine reduced the activity whereas reversing the charge of both arginines abolished activity, suggesting that the charge of these amino acids is important. Finally, mutating Leu290 to proline abolishes activity almost completely. Together with Arg288 and Tyr289, these amino acids are part of the helical region that melts upon DNA binding and lie very

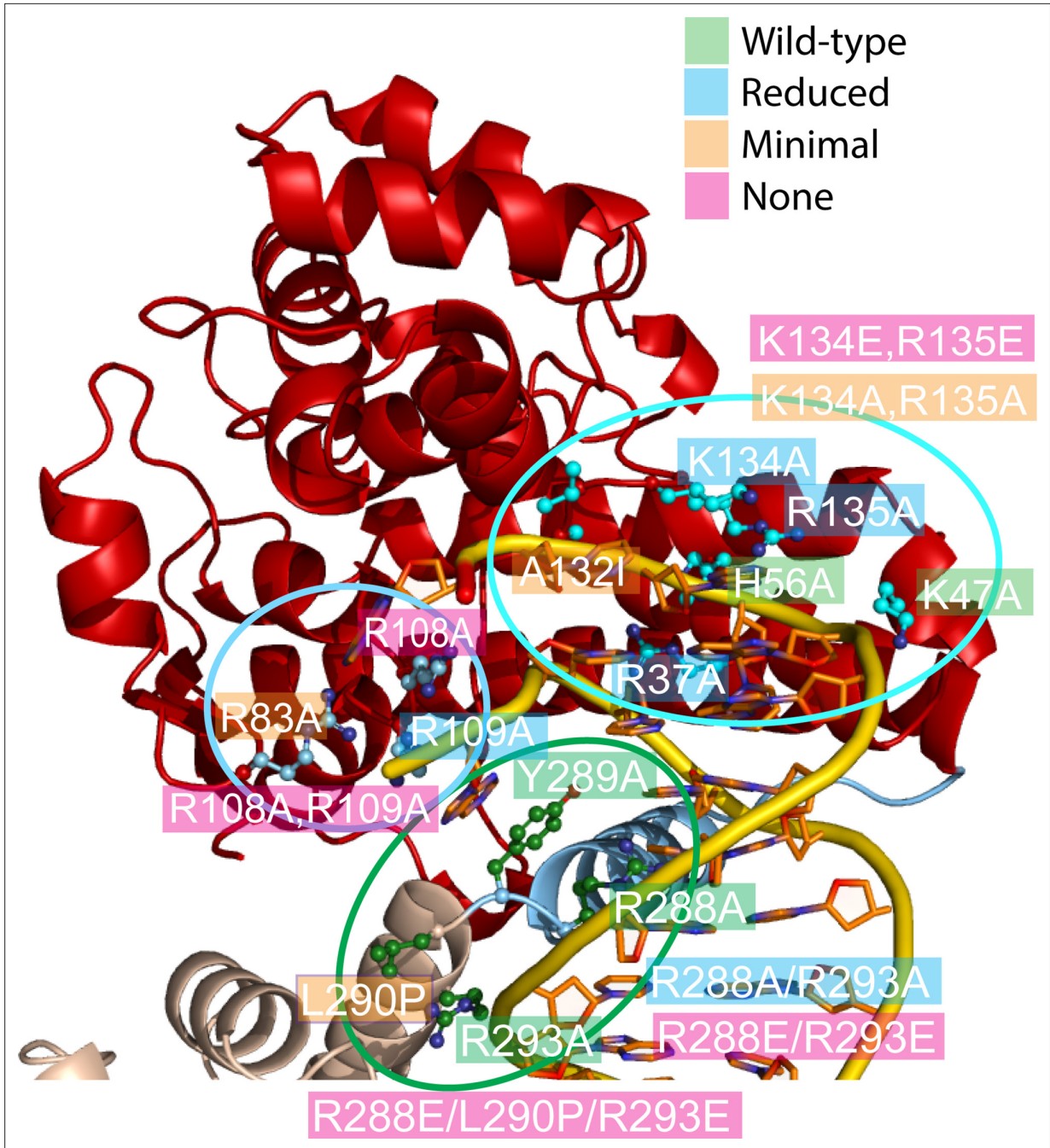

**Figure 4.** Mutagenesis supports the role of different topoisomerase V residues in DNA binding and conformational changes. Site-directed mutagenesis of the protein (*Table 5*) was used to probe the role of various residues around three regions (delineated by ellipses). The figure shows the mutated residues colored by results (green: wild-type level of activity, blue: reduced activity, orange: minimal activity, and pink: no activity). In many instances single mutants have a modest effect, but combination of them have more dramatic results. Arg108 is essential for activity and Arg83, Ala132, and Leu290 show very reduced activity. Arg108 is near the active site and may participate in catalysis, whereas Leu290 is part of the linker region that changes conformation upon DNA binding. Single mutations of positively charged residues facing DNA tend to be benign, but combinations or charge reversal led to reduced activity.

The online version of this article includes the following source data and figure supplement(s) for figure 4:

**Figure supplement 1.** DNA relaxation assay for the topoisomerase V mutants analyzed.

**Figure supplement 1—source data 1.** DNA relaxation assay for topoisomerase V mutants analyzed.

**Figure supplement 1—source data 2.** DNA relaxation assay for topoisomerase V mutants analyzed.

*Figure 4 continued on next page*

*Figure 4 continued*

**Figure supplement 1—source data 3.** DNA relaxation assay for topoisomerase V mutants analyzed.

**Figure supplement 1—source data 4.** DNA relaxation assay for topoisomerase V mutants analyzed.

close to the phosphate backbone. The Leu290Pro was designed to interfere with helix formation and its lack of activity suggests that interfering with the structure inactivates the enzyme. Not surprising, an Arg288Glu/Leu290Pro/Arg293Glu triple mutant has no activity. The results show that while amino acids that approach the DNA are important, the ability of the linker helix to change conformation has the most dramatic effect. The results confirm the importance of the linker helix in the overall conformation of the enzyme.

### Active site

The residues forming the active site were identified from the structure of a 61 kDa fragment (*Taneja et al., 2006*) and later confirmed by site-directed mutagenesis (*Rajan et al., 2014*). Aside from Tyr226, the active site tyrosine, five residues were identified as playing a role in the cleavage/religation reaction: Arg131, Arg144, His200, Glu215, and Lys218. The structure of the complex shows that aside from Lys218 all these residues are in the vicinity of the DNA (*Figure 5*). Based on the structure, the putative scissile phosphate, termed here P0, would correspond to the last phosphate in the oligonucleotide but is not present in the structures, as the 3' end of the oligonucleotides is dephosphorylated. In all oligonucleotides studied, the 3' end of the last nucleotide is in the same general location, near the

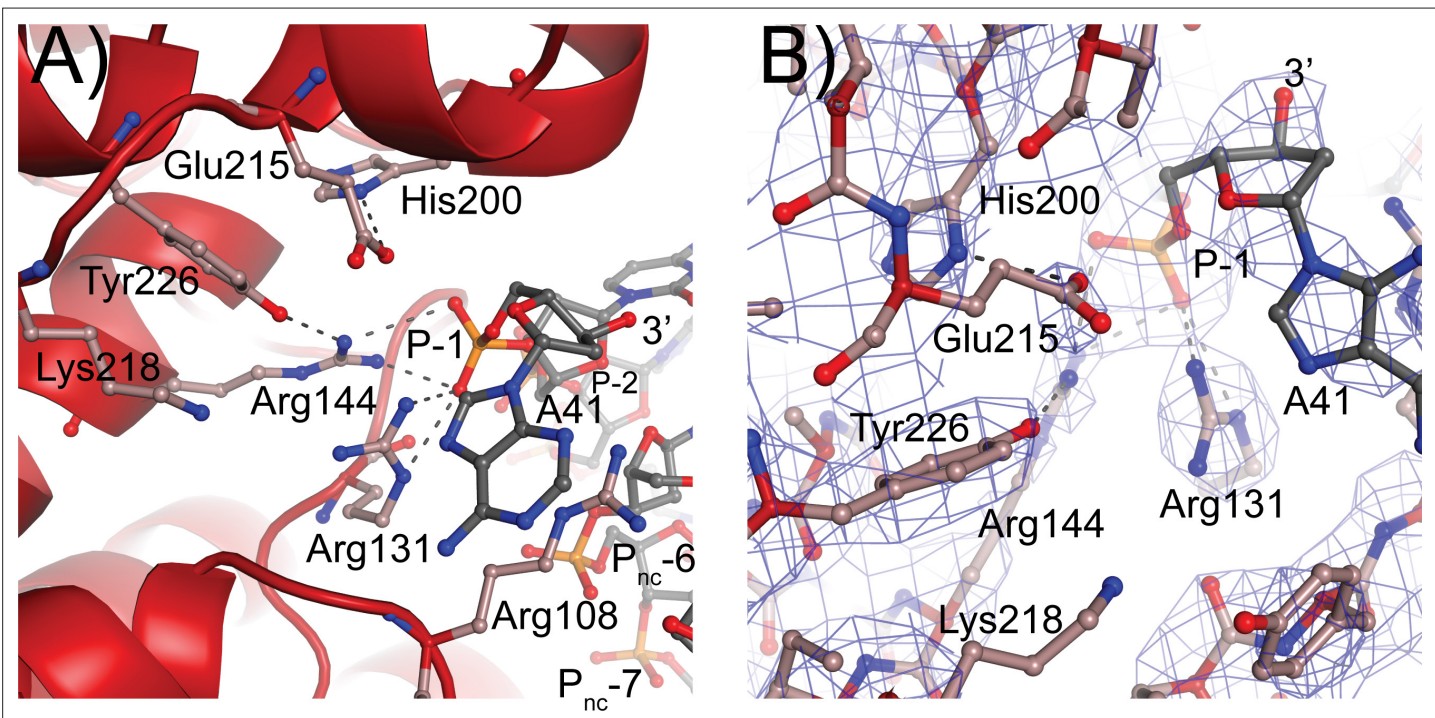

**Figure 5.** Topoisomerase V active site. (**A**) Diagram of the active site in the topoisomerase V with 40-bp symmetric DNA complex. The diagram shows the side chains that have been implicated in the cleavage/religation reaction. The DNA approaches the active site and comes close to the active site tyrosine (Tyr226). The preceding phosphate (P-1) is contacted by three arginines, Arg108, Arg131, and Arg144. His200 is too far away from the phosphate backbone, but is hydrogen bonded to Glu215. The latter is in a suitable position to contact the DNA phosphate backbone. Lys218 is in a position where it could contact the phosphate backbone during cleavage/religation. Arg108, which was identified as essential, lies between the two DNA strands and contacts both the P-1 phosphate on the cleaved strand and the $P_{nc}$-6 phosphate on the noncleaved strand. Due to the melting of the DNA the cleaved and noncleaved strands follow different paths resulting in the phosphate five bases away in the noncleaved strand ($P_{nc}$-6) facing the P-1 phosphate in the cleaved strand. (**B**) Simulated annealing omit map of the topoisomerase V with 40-bp symmetric DNA complex structure. The diagram presents another view of the active site as well as the omit electron density around that region at the 1 σ level.

The online version of this article includes the following figure supplement(s) for figure 5:

**Figure supplement 1.** Electron density maps of the topoisomerase V in complex with 39-bp symmetric DNA.

tyrosine, but pointing away from it. A simple rotation around one of the bonds would bring it near the active site tyrosine. As the protein opens to the solvent in this region, there are few obstacles to hinder the rotation and allow the phosphate backbone to enter completely into the active site; the nucleo-base could easily move into the solvent region. Arg131 and Arg144 both contact the P-1 phosphate, the phosphate group immediately 5′ of the scissile one (*Figure 5*). Their orientation is such that they could also contact the P0 phosphate if it was present as they are in the region between the two phosphates. His200 and Glu215 are hydrogen bonded to each other. In the complex structure, they are too far to contact the DNA directly, but Glu215 could contact the P0 phosphate with minimal side chain rearrangements. Finally, Lys218 is not making contacts with the DNA but is close enough to Tyr226 that it could contact the P0 phosphate. Mutagenesis of additional residues indicates that Arg108 is essential. This arginine is wedged between the two DNA strands and is in position to contact both the P-1 phosphate and a phosphate group of the noncleaved DNA strand, which due to the melting of the DNA the latter would correspond to the complementary nucleotide five bases away and labeled as the $P_{nc}$-6 phosphate in *Figure 5*. Its position in the complex and the mutagenesis results suggest that Arg108 is critical for activity as it helps position the cleaved strand for catalysis while holding to noncleaved strand and perhaps also aiding in getting the two strands closer together. In addition, Arg83 was also found to be important as changing it to alanine has minimal activity. Arg83 is also in a position to interact with the phosphate backbone of the noncleaved strand, suggesting that both Arg83 and Arg108 are important residues for activity by interacting with the noncleaved strand, probably by ensuring that it remains in place during catalysis. Thus, all residues that have been involved in cleavage and religation are positioned in a manner consistent with their previously assigned roles (*Rajan et al., 2014*) and additional important residues have been identified.

## Discussion

The general outlines of the DNA-binding mode and relaxation mechanism is understood for many topoisomerases (*Bush et al., 2015*; *Corbett and Berger, 2004*), but this is not the case for type IC enzymes. In the latter case, the absence of information on the way they recognize and bind DNA has limited our understanding of the mechanism of this topoisomerase subtype. In addition, the lack of sequence or structural similarities between topoisomerase V and other topoisomerases meant that it was not possible to deduce the DNA-binding mechanism based only on similarities. The presence of multiple $(HhH)_2$ domains as well as biochemical information suggested that these domains were involved in DNA binding (*Belova et al., 2002*; *Rajan et al., 2010*), but it was not clear how tandem domains recognize and bind DNA. The structures presented here show the way topoisomerase V interacts with DNA, both through the $(HhH)_2$ domains and the topoisomerase domain. DNA binding by the $(HhH)_2$ domains is unusual and, in a way, not previously observed for other proteins. The tandem $(HhH)_2$ domains wrap around the DNA forming a loop-like structure that encircles the DNA. Typically, HhH repeats are found as single repeats or forming $(HhH)_2$ domains (*Shao and Grishin, 2000*), not as tandem arrangements. Some proteins that include $(HhH)_2$ domains dimerize to have two $(HhH)_2$ domains next to each other, for example XPF endonucleases (*Nowotny and Gaur, 2016*), but the $(HhH)_2$ domains are not arranged in a tandem configuration in the protein. For this reason, even though HhH repeats and $(HhH)_2$ domains are well studied, it was not possible to model DNA binding by topoisomerase V based on known structures. Unlike other $(HhH)_2$ domain-containing proteins, where the domains are involved in DNA binding and repair, in topoisomerase V some $(HhH)_2$ domains surround the DNA but do not contact it, others contact it, and only three out of twelve are directly involved in repair. When comparing individual $(HhH)_2$ domains with repair enzymes that interact with DNA the interaction observed in topoisomerase V is different from the one observed in other $(HhH)_2$ domain complexes with DNA. It is not possible to establish whether the topoisomerase V repeats directly involved in DNA repair bind DNA similar to the way other $(HhH)_2$ domain-containing DNA repair enzymes do. The $(HhH)_2$ domains that contain the AP/dRP lyase activity do not engage directly with the DNA abasic sites in the structure. For this reason, the question on how topoisomerase V recognizes DNA lesions is not completely answered by the current structure. To understand the way the enzyme recognizes and cleaves abasic sites additional structural information on the interactions of the repair domains and an abasic site is needed.

The overall DNA-binding mode by topoisomerase V is unusual. The $(HhH)_2$ domains surround the DNA loosely covering almost four helical turns. Topoisomerase V clamps around DNA by having two

sets of tandem repeats of (HhH)$_2$ domains that follow the path of DNA in opposite directions. The region between repeats 7 and 8 serves as the turning point to permit the repeats to change direction and encircle the DNA. The Topo-97(ΔRS2) fragment is missing the last two (HhH)$_2$ domains plus a few amino acids of unknown structure at the C-terminus. Based on the structures and the modeling studies described, it is likely that the last two (HhH)$_2$ domains continue the same path as the previous three and interact with the topoisomerase domain and completely enclose the DNA forming two full loops (*Figure 2—figure supplement 4*). Given the length of DNA covered, there are few direct interactions with the phosphate backbone and very few with the nucleobases. Instead, it appears that the (HhH)$_2$ domains create an enclosed positively charged track or groove where the DNA can travel. Other proteins that use tandem repeats of DNA-binding domains do not make a turn to change direction while binding DNA. Instead, they wrap around the DNA in a single direction. For example, zinc finger containing proteins such as Zif268 or Gli (*Pavletich and Pabo, 1991*; *Pavletich and Pabo, 1993*) and TAL effector proteins binding domains (*Mak et al., 2012*) interact with the DNA by having tandem repeats that wrap around the DNA, but they extend along the DNA, not turning to form a loop-like structure that surrounds it.

DNA binding by topoisomerase V does not require all (HhH)$_2$ domains, but its activity is enhanced when more (HhH)$_2$ domains are present, suggesting that even a few domains are capable of binding DNA (*Belova et al., 2002*; *Rajan et al., 2010*). Surrounding DNA is not necessary for activity, but it clearly enhances it. Furthermore, it has been shown that the topoisomerase V (HhH)$_2$ domains can be used to enhance processivity of other enzymes (*Pavlov et al., 2002*). Thus, the (HhH)$_2$ domains may serve a similar role as sliding clamps that surround DNA (*Hedglin et al., 2013*), which enhance processivity by keeping polymerases closely associated with DNA while allowing movement along the DNA. Unlike sliding clamps, which form a ring around DNA and need a clamp loader to assemble, the topoisomerase V (HhH)$_2$ domains surround the DNA in a more extended manner and load into DNA without assistance.

The active site of topoisomerase V had not been observed in an accessible conformation before. All other known structures (*Rajan et al., 2016*; *Rajan et al., 2013*; *Rajan et al., 2010*; *Taneja et al., 2006*), aside from the structure of the isolated topoisomerase domain (*Rajan et al., 2010*), showed the active site obstructed by the (HhH)$_2$ domains. Whereas it was clear that the (HhH)$_2$ domains had to move to allow access to the active site, it was not clear how this is accomplished. The structure of the symmetric complex with DNA shows that the topoisomerase domain moves relative to the (HhH)$_2$ domains through a single conformational change, a kink in the helix linking the first (HhH)$_2$ domain and the topoisomerase domain. The domains themselves move as rigid bodies; the only change occurs in the linker helix. This simple mechanism allows the domains to separate and exposes the active site. In addition, the linker helix interacts with the DNA directly, suggesting that the conformational change could be triggered by protein DNA interactions. Single mutations in the linker helix did not disrupt activity, aside from one, Leu290Pro. Arginines 288 and 293 face the major groove and single mutations to alanine had no effect. A double mutant with both arginines changed to alanines had reduced activity whereas reversing the charge of these residues, Arg288Glu/Arg293Glu, completely abolished activity. This suggest that the overall charge character of the region is important for protein/DNA interactions as the positively charge side chains may facilitate the approach of the DNA to the linker helix. Leu290Pro was designed to disrupt the helical structure of the linker helix. It is positioned at the end of the break point in the linker helix. The presence of the proline abolished activity, supporting the notion that the structure and plasticity of the linker helix is very important for activity. One possible explanation for the loss of activity is that the proline prevents the formation of the linker helix in the closed or DNA-free form, thus affecting the ability of the protein to recognize and bind DNA and attain the proper conformation that allows binding of the DNA by the (HhH)$_2$ domains and orienting the topoisomerase domain properly for catalytic activity.

The movement of the domains also exposes a large, positively charged patch in the topoisomerase domain where the DNA enters the active site. In order to enter the active site, the DNA bends at two positions. There is small bend adjacent to the abasic site and a much larger bend where the DNA enters the topoisomerase domain. It is unlikely that the abasic site-induced bending reflects a feature of DNA binding and recognition as nondamaged DNA is easily relaxed by topoisomerase V. Surprisingly, the DNA is highly bent where it enters the topoisomerase domain and the bending has an unusual consequence, the double stranded DNA melts to allow only one strand to enter the

active site. The two DNA strands separate, breaking the base pairing between them. One strand, the noncleaved one, is anchored to the protein by clamping on a nucleobase. The other strand enters the active site and approaches the active site tyrosine. Both DNA strands exit the protein almost parallel to each other, suggesting that base pairing would resume once the DNA exits the protein. Single mutations of positively charged amino acids leading to the positively charged patch have, in general, a minor effect. Arg37, Lys47, His56, Arg134, and Arg135 were mutated to alanine with almost no reduction in relaxation activity. Interestingly, a double mutant where Lys134 and Arg135 were mutated to alanines had minimal activity, but when these amino acids were mutated to glutamate, reversing the charge of the region, activity was completely lost. These results suggest that single changes do not alter enough of the charge character of the region to significantly reduce the activity, but that more extensive changes do affect the activity and, finally, that the positively charge character of the region is crucial for activity. The mutation Ala132Ile was designed to probe the structure of a region that is in intimate contact with the cleaved DNA strand through protein backbone interactions. This mutant showed significant loss of activity, even though the side chain does not face the DNA. Ala132 faces the interior of the protein and it is likely that the bulkier isoleucine side chain affected the structure of the region and hindered protein/DNA interactions, supporting the notion that the structure of this region is important for protein/DNA interactions. Mutations of charged amino acids in contact with the noncleaved DNA strand had a larger effect on activity. Arg109Ala did not show any change, but Arg83Ala and Arg108Ala show significant decrease in activity. In particular Arg108Ala abolishes activity completely. Their position in the complex suggests that Arg83 and Arg108 may interact with the noncleaved strand and hold it in place during DNA relaxation activity. It is interesting to note that Arg108Ala shows no activity and is near the active site, suggesting that this amino acid may also play a role in catalysis that was not noticed before. Not surprising, the double mutant of Arg108 and Arg109 to alanines show no activity.

The residues forming the topoisomerase active site were recognized based on structural and biochemical observations (*Rajan et al., 2014*; *Taneja et al., 2006*). The complex structure confirms that Arg131, Arg144, His200, Glu215, Lys218, and Tyr226 are involved in interacting with DNA and helps understand their role in the DNA cleavage/religation reaction. The two arginines, Arg131 and Arg144, interact directly with the phosphate group adjacent to the scissile bond (P-1) and are likely to interact also with the scissile bond phosphate (P0). Arg144 also forms a hydrogen bond to the OH group of the active site tyrosine. Mutations in either arginine led to very reduced activity (*Rajan et al., 2014*), suggesting an important role for these residues in the reaction. It was suggested (*Rajan et al., 2014*) that these arginines play a role in transition state stabilization, which would be broadly consistent with the observations from the complex structure. Glu215, an unusual residue due to its negative charge, comes close to the phosphate backbone and hydrogen bonds to His200. It was observed before that Glu215 reduces the binding affinity of the protein for DNA, which would be consistent with a negatively charged residue approaching the negatively charged phosphate backbone (*Rajan et al., 2014*). In the complex structure, His200 does not seem to interact directly with DNA so its role in the cleavage/religation reaction is not clear. Finally, the role of Lys218 is also not clear from the structure, as it is not observed in a position to interact with DNA, probably due to the absence of P0. It appears that it could interact with the phosphotyrosine intermediate helping stabilize it after cleavage; mutational studies show that the lysine is essential (*Rajan et al., 2014*). Finally, the finding that Arg108 is essential for activity suggests that it may play a role in catalysis, either through a direct role in cleavage/religation or stabilizing the DNA, that had not been recognized before. The combined structural and biochemical studies do suggest that the mechanism of cleavage and religation is different from the one employed by type IB enzymes. In type IB enzymes the histidine in the catalytic pentad interacts directly with DNA and the lysine plays a role in proton transfer. This does not appear to be the case in type IC topoisomerases. The presence of the glutamate in close contact with the phosphate backbone as well as the potential different role of His200 and Lys218 suggest that a distinct catalytic strategy is employed by topoisomerase V. Additional structural studies to capture covalent intermediates are needed to understand the cleavage/religation reaction.

Both type IB and IC topoisomerases use a swiveling mechanism to relax DNA. Structures of type IB enzymes in complex with DNA (*Perry et al., 2006*; *Redinbo et al., 1998*) show that the DNA remains double helical and largely unbent and is surrounded by the protein (*Figure 6*). Cleavage and formation of a transient phosphotyrosine intermediate allow the intact strand to rotate until the broken free

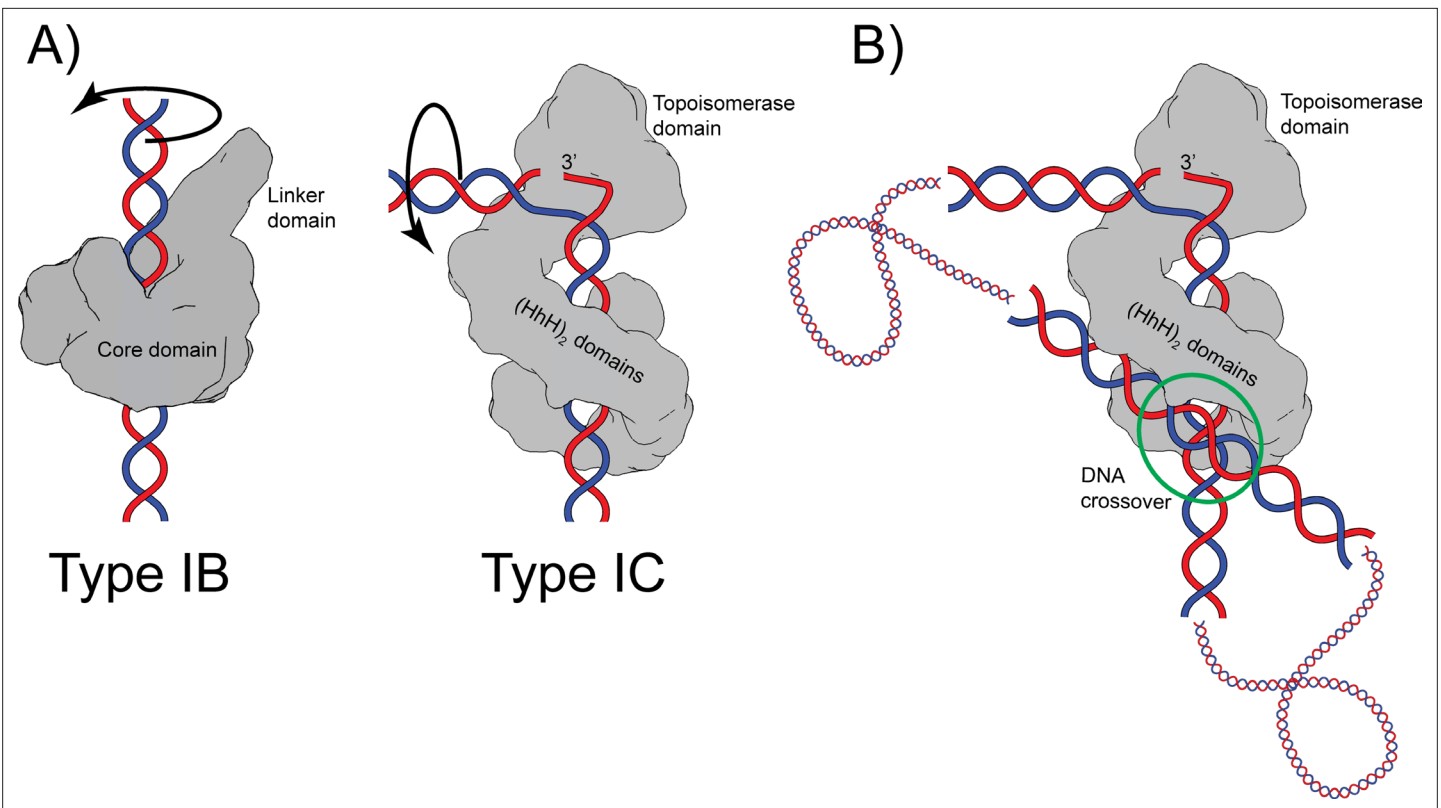

**Figure 6.** Proposed mechanism of relaxation for type IB and IC topoisomerases. (**A**) Cartoons of the proposed relaxation mechanism for type IB and IC enzymes, illustrated by a human topoisomerase complex and topoisomerase V, respectively. The proteins are shown as a surface based on their structure whereas the DNA is shown in cartoon form. (Left) Type IB enzymes relax DNA by enclosing it and cleaving one strand, allowing for strand rotation or swiveling. The DNA is not bent or distorted. (Right) Type IC topoisomerases do not surround the DNA around the active site region. They surround the DNA using tandem (HhH)$_2$ domains, which appear to serve as a processivity factor. The DNA is highly bent around the active site region and this bending melts the two strands, allowing one of them to enter the active site. The way the two proteins interact with DNA is very different in both types. In addition, their DNA cleavage and religation mechanism appear to be different. In both cases, interactions between the protein and DNA create friction, which modulates the rate of the reaction (*Koster et al., 2005*; *Taneja et al., 2007*). Supercoiling of the DNA creates torque, which drives the reaction. The type IB diagram was drawn based on the structure of human topoisomerase I in complex with DNA (PDB 1K4T) (*Staker et al., 2002*). (**B**) Cartoon depicting the possible binding of topoisomerase V to two DNA strands at a DNA crossover point (green circle). Binding at a DNA crossover point would serve as a way to sense overall DNA topology. Two different binding sites are required, which in this case would correspond to the two sites observed in the complex structures. It has been suggested that type IB enzymes sense overall DNA topology by binding to crossover points (*Madden et al., 1995*; *Zechiedrich and Osheroff, 1990*), which would represent another possible commonality between the mechanism of type IB and IC topoisomerases.

DNA strand is recaptured and ligated. Interaction of the DNA with the protein causes friction, which modulates the reaction (*Koster et al., 2005*). Single molecule experiments on topoisomerase V show a similar overall mechanism where friction also plays a role (*Taneja et al., 2007*), suggesting that type IB and IC enzymes employ the same general strategy. Given these parallel strategies, the presence of a sharp bend and a single stranded region in the complex of topoisomerase V was unexpected as type IB topoisomerases do not bend or melt the DNA (*Perry et al., 2006*; *Redinbo et al., 1998*). Single stranded DNA recognition and binding is part of the type IA mechanism, but these enzymes work by an enzyme-bridged strand passage mechanism, which is fundamentally different from the swiveling mechanism employed by type IB and IC enzymes. These observations suggest that despite the apparent similarities, type IC enzymes employ a different relaxation strategy. Unlike type IB enzymes, type IC molecules bend the DNA to create a single stranded region that is likely to facilitate swiveling by freeing the two DNA strands around the cleavage site. It is not clear whether the rotation of the strands involves only rotation of the strands or also movement of the topoisomerase domain; it is possible that the topoisomerase domain moves as the strands rotate (*Figure 6A*). Similar to type IB enzymes, the broken strand is captured after swiveling around the other strand. Finally, it is interesting

to note that whereas type IB enzymes surround the DNA during the reaction, type IC enzymes do not appear to do so. The (HhH)$_2$ domains surround the DNA, but their role seems to be to act as a processivity factor as these domains are not required for DNA relaxation (*Rajan et al., 2010*). Thus, while there are some overall similarities between the two subtypes, there are important differences that suggest that topoisomerase V employs a completely different relaxation mechanism.

It has been observed previously that type IB enzymes bind at crossover points in supercoiled DNA and that this may provide a mechanism for overall topology sensing (*Madden et al., 1995*; *Zechiedrich and Osheroff, 1990*). Two independent DNA-binding sites in the protein are needed to bind two strands of DNA at a crossover point. Crossover binding provides a facile way to recognize the overall topology of DNA as, in general, only supercoiled DNA, either positively or negatively supercoiled, will form crossovers. Viral type IB topoisomerases have been observed to synapse DNA strands (*Moreno-Herrero et al., 2005*; *Shuman et al., 1997*), again suggesting the presence of more than one DNA-binding site. Whereas there are no structures of any topoisomerase binding two DNA strands simultaneously, there are electron (*Shuman et al., 1997*; *Zechiedrich and Osheroff, 1990*) and atomic force microscopy (*Moreno-Herrero et al., 2005*) studies that support the notion of multiple binding sites as well as a structure of a bacterial type IB enzyme bound to a secondary DNA site (*Patel et al., 2010*). The latter structure supports the notion that type IB enzymes can bind two DNA molecules simultaneously and independently. It is possible that topoisomerase V, as has been suggested for type IB enzymes, recognizes the overall topology of DNA by preferentially binding at crossover points. The structure of the asymmetric complex of topoisomerase V with DNA shows the presence of a second DNA-binding site that involves the external surface of the (HhH)$_2$ domains and that does not interfere with binding of the strand that enters the active site (*Figure 1*). *Figure 6B* shows a cartoon of topoisomerase V bound to two DNA segments. It shows that it could be possible for topoisomerase V to recognize the overall DNA topology by simultaneously binding two DNA strands. This would make type IC topoisomerases share with type IB enzymes not only a similar relaxation mechanism, controlled strand rotation, but possibly an overall topology sensing mechanism. Whether type IB and/or IC enzymes use a crossover point binding mechanism to sense overall topology remains to be proven, but it adds to the mechanistic similarities between the two types of topoisomerases despite their very different structures and supports the notion of convergent evolution to a common mechanism of DNA topology sensing and relaxation.

The structures presented here add significantly to our understanding of type IC enzymes and their mechanism. The structures show that an important role of the (HhH)$_2$ domains is to surround DNA and, in this manner, serve as a processivity factor. In this way, they keep the enzyme bound to DNA, probably allowing it to travel along DNA. It is not clear from the structures how DNA lesions are recognized, but if the protein slides along the DNA it could scan it for lesions using the (HhH)$_2$ domains containing the repair active sites. The topoisomerase domain is likely to be obstructed in the absence of DNA/protein interactions, but a simple conformational change around a linker helix exposes the active site once the protein is bound to DNA. The DNA bends tightly as it interacts with the protein around the active site region, leading to single stranded DNA formation, which may help facilitate strand rotation. Finally, the catalytic mechanism of DNA cleavage and religation appears to be different from the one employed by other topoisomerases, despite some very general similarities. All these observations indicate that topoisomerase V is a multifaceted enzyme that encompasses in the same polypeptide a novel topoisomerase domain, DNA repair domains, and a DNA clamp. Thus, type IC enzymes are fundamentally different from all other topoisomerases. The existing structural, biochemical, and biophysical data help to establish firmly type IC topoisomerases as a completely different topoisomerase subtype with few sequence, structural, or mechanistic similarities to all the other subtypes.

# Materials and methods
## Protein purification

A fragment of Topo-V corresponding to residues 1–854 (Topo-97) with Lys809, Arg820, Arg831, Arg835, Arg846, and Arg851 mutated to alanine to remove the second AP lyase site (Topo-97(ΔRS2)) has been previously described (*Rajan et al., 2016*). For protein purification, Topo-97(ΔRS2) was transformed into *E. coli* BL21 Rosetta (DE3) cells. Protein induction and purification were done according to

previously described protocols (*Rajan et al., 2013*; *Rajan et al., 2010*). Pure protein was concentrated to 55.6 mg/ml and stored in 50 mM Tris pH 8, 500 mM NaCl, and 1 mM dithiothreitol (DTT). Protein purity was assessed by polyacrylamide gel electrophoresis.

## Oligonucleotide

Oligonucleotides were purchased from Integrated DNA Technologies (IDT, Coralville IA) at 250 nmol. The sequences of the oligonucleotides used are shown below.

| Name | Sequence |
|---|---|
| 38 bp asymmetric* | 5'– TGCCTGCACGAABTAAGCAATTCGTAATCATGGTGCGCCA –3'<br>3'– GTACGGACGTGCTTCATTCGTTAAGCATTAGTACCACGCG –5' |
| 38 bp symmetric | 5'– TGCCTGCACGAABTAAGCATGCTTACTTCGTGCAGGCACA –3'<br>3'– GTACGGACGTGCTTCATTCGTACGAATBAAGCACGTCCGT –5' |
| 39 bp symmetric | 5'– TGCCTGCACGAABTAAGCATTGCTTACTTCGTGCAGGCACA –3'<br>3'– GTACGGACGTGCTTCATTCGTAACGAATBAAGCACGTCCGT –5' |
| 40 bp symmetric | 5'– TGCCTGCACGAABTAAGCATATGCTTACTTCGTGCAGGCACA –3'<br>3'– GTACGGACGTGCTTCATTCGTATACGAATBAAGCACGTCCGT –5' |

*For the high-resolution native dataset the abasic site was replaced with a complementary G.
The blue shows shows the position of the twofold axis in the crystal. Only the 39-bp symmetric oligonucleotide is symmetric in the crystal. The abasic sites are denoted by a B. They correspond to a tetrahydrofuran abasic site to mimic an apurinic/apyrimidinic (AP) site.

Individual oligonucleotides were resuspended to 1 mM in water. For annealing, complementary oligonucleotides were mixed at an equimolar ratio, heated to 85°C for 2.5 min, cooled down to 5°C below their calculated melting temperature for 5 min, and then transferred to ice for at least 10 min. Annealed oligonucleotides were used directly in crystallization experiments.

**Table 6.** Primers used for site-directed mutagenesis.

| Mutant | Primer |
|---|---|
| Leu290Pro | Forward<br>5'-GACATCATGAGAAGGTAT**CCT**GAGCAGCGGATC-3'<br>Reverse<br>5'-GATCCGCTGCTC**AGG**ATACCTTCTCATGATGTC-3' |
| Arg109Ala | Forward<br>5'-GATCGTGTACAGG**GCA**GGCTGGAGGGCGATC-3'<br>Reverse<br>5'-CGCCCTCCAGCC**TGC**CCTGTACACGATC-3' |
| Arg288Ala, Arg293Ala | Forward<br>5'-GCGACATCATGAGA**GCG**TATCTCGAGCAG**GCG**ATCGTCGAGTGT-3'<br>Reverse<br>5'-ACACTCGACGAT**CGC**CTGCTCGAGATA**CGC**TCTCATGATGTCGC-3' |
| Arg288Glu, Arg293Glu* | Forward<br>5'-GCGACATCATGAGA**GAG**TATCTCGAGCAG**GAG**ATCGTCGAGTGT-3'<br>Reverse<br>5'-ACACTCGACGAT**CTC**CTGCTCGAGATA**CTC**TCTCATGATGTCGC-3' |
| Arg288Glu, Leu290Pro, Arg293Glu† | Forward<br>5'-CATGAGA**GAG**TAT**CCT**GAGCAG**GAG**ATCGTC-3'<br>Reverse<br>5'-GACGAT**CTC**CTGCTC**AGG**ATA**CTC**TCTCATG-3' |
| Lys134Glu, Arg135Glu | Forward<br>5'-AGAGGTGCGTGCCGTG**GAGGAG**AACCCGCTCCAACCGG-3'<br>Reverse<br>5'-CCGGTTGGAGCGGGTT**CTCCTC**CACGGCACGCACCTCT-3' |

*The double mutant Arg288Glu, Arg293Glu DNA backbone was used in mutagenesis PCR to get the Arg288Ala, Arg293Ala mutant.
†The double mutant Arg288Glu, Arg293Glu DNA backbone was used in mutagenesis PCR to get the Arg288Glu, Leu290Pro, Arg293Glu triple mutant.

## Site-directed mutagenesis

Arg37, Lys47, His56, Arg83, Arg108, Arg109, Lys134, Arg135, Arg288, Tyr289, Leu290, and Arg293 were individually mutated to the residues listed in *Table 5* on the pET-21b(+)T97 expression plasmid (*Rajan et al., 2016*). The pET-21b(+)T97 plasmid carries the coding sequence for the 97 kDa amino-terminal fragment of the Topo-V (Topo-97) that extends to residue 854. A total of 19 mutants were made either by site-directed mutagenesis (6 mutants) or by complete gene synthesis by GenScript (Jiangsu, China) (13 mutants) (*Table 5*). Mutations generated by site-directed mutagenesis were introduced with complementary mutagenic primers (Integrated DNA Technologies, Coralville, IA) (*Table 6*) into the pET-21b(+)T97. KOD Hot Start DNA polymerase (MilliporeSigma Novagen) was used for site-directed mutagenesis following the manufacturer suggested protocol except for Leu290Pro. In the Leu290Pro case, the touchdown PCR method using iProof High-Fidelity DNA polymerase, GC Master Mix with 3% dimethyl sulfoxide (Bio-Rad, CA) was used. Sanger sequencing was performed on the site-directed mutant plasmids to determine both that the intended mutations were introduced correctly into the plasmid and that any unintended mutations were not introduced. The synthesized mutants were made using the pET-21b(+)T97 plasmid as a backbone and sequenced by Genscript to verify the sequence.

## Mutant proteins purification

For the activity assays, wild-type and mutant Topo-97 proteins were transformed into *E. coli* BL21 Rosetta (DE3) cells. Transformants were grown at 37°C in Luria-Bertani media containing 100 µg/ml ampicillin and 100 µg/ml chloramphenicol until they reached an optical density $A_{600}$ of 0.6–0.8. Subsequently the cells were cooled down on ice and induced with 1 mM isopropyl β-D-1-thiogalactopyranoside for 14–16 hr at 16°C. Cells were then spun down at 4000 revolutions per minute (rpm) in a swinging bucket rotor (Eppendorf 5810 Centrifuge); cell pellets were flash frozen in liquid nitrogen and stored at −80°C until use. After thawing the pellets, cells were resuspended in Topo-V resuspension buffer (50 mM Tris–HCl pH 8, 500 mM NaCl, 1 mM ethylenediaminetetraacetic acid [EDTA] pH 8, 1 mM DTT) with 1 mM benzamidine, 1 mM phenylmethylsulfonyl fluoride, 0.1% Brij 58, and one dissolved protease inhibitor, EDTA-free mini tablet (Thermo Scientific Pierce). After resuspension, cells were lysed by sonication and the lysate was clarified by two centrifugation spins at 35,000 rpm in a Ti-70 rotor (Beckman Coulter) in an Optima XE-90 ultracentrifuge (Beckman Coulter) at 4°C. After the first ultracentrifugation spin, clarified supernatant was heated to 75°C for 25 min to help further purify the extreme thermophilic Topo-V from any contaminating proteins. Subsequently, the heated supernatant was spun to further clarify the supernatant. Following the second ultra-centrifugation spin the supernatant was passed through a 0.2-µm filtration unit and the protein was concentrated with a 50 K MWCO concentrator (Thermo Scientific Pierce Protein Concentrators PES, 50 K MWCO) to concentrate Topo-97 and to filter out any residual contaminating proteins. Purified proteins were flash frozen and stored at −80°C. During protein purification, samples were taken for each step and analyzed by Coomassie-stained sodium dodecyl sulfate–polyacrylamide gel electrophoresis (SDS–PAGE) gels to determine purity.

## DNA relaxation assays

DNA relaxation assays were carried out as described previously (*Rajan et al., 2014*). Briefly, 0.15, 1.5, and 3.5 µg of the enzyme were incubated with 306 ng of negatively supercoiled pUC19 plasmid in a 15 µl of reaction containing 50 mM sodium acetate pH 5 at 65°C, 30 mM NaCl, and 0.2 mM EDTA for 30 min at 65°C. The reaction was terminated by moving to ice and adding 1.3% SDS. The products were resolved on a 1% agarose gel and visualized by ethidium bromide staining.

## Dynamic light scattering

DLS measurements were done with a Punk (Unchained Labs) DLS instrument to determine the hydrated radius and calculated molecular weight of the free protein and complexes with DNA. All measurements were done using the manufacturer's recommended procedure. Measurements were taken of the 112 kDa full-length wild-type topoisomerase V and the Topo-97(ΔRS2) fragment alone and in complex with 38 bp asymmetric and 40 bp symmetric abasic DNA. Initial DLS measurements of the free protein or the protein/DNA complex showed aggregation. Good measurements were obtained once experimental preparation procedures were optimized as follows: the protein was

incubated (50 mM sodium acetate pH 5 at 65°C, 30 mM NaCl, 1 mM MgCl$_2$) with the substrate at 65°C for 15 min in 30 µl. After 15 min a prewarmed buffer (50 mM sodium acetate pH 5 at 65°C, 750 mM NaCl, 1 mM MgCl$_2$) was added to each of the six reaction tubes to bring the salt up to 350 mM NaCl. The samples were then incubated with the higher salt buffer at 50°C for 15 min followed by incubation at room temperature for 2 days. The following samples were measured: Topo-97(ΔRS2) alone and in complex with 38-bp asymmetric or 40-bp symmetric DNA and full-length topoisomerase V alone and with 40-bp asymmetric DNA. The sample concentrations used were: 52.5 µM Topo-97(ΔRS2) mixed with 17.5 µM 38 or 40 bp DNA, free Topo-97(ΔRS2) at 30.8, 17.5, and 11.75 µM full-length topoisomerase V with 5.85 µM 40 bp asymmetric DNA, and 11.6 µM free full-length topoisomerase V. Despite the optimization procedure, the samples using the full-length protein showed aggregation, giving unreliable estimates. The hydrated diameter of the models was calculated using the program HullRad (*Fleming and Fleming, 2018*). Comparison between the observed and calculated diameters served to distinguish between models.

## Crystallization, data collection, and structure determination

For all crystallization experiments, Topo-97(ΔRS2) was mixed with the corresponding oligonucleotide using a stoichiometric ratio of 1.25:1 DNA to protein in DNA-binding buffer (50 mM sodium acetate pH 5.0 at 65°C, 30 mM sodium chloride, 1 mM magnesium chloride). Reactions were incubated for 30 min at 65°C. For the asymmetric DNA experiments, molar ratios of 40:32 and 60:48 µM DNA to protein were used whereas for the symmetric DNA experiments a ratio of 60:48 µM DNA to protein yielded the best crystals.

All crystallizations were done by vapor diffusion at 30°C in a hanging drop setup. Topo-97(ΔRS2) with asymmetric DNA crystals typically started to appear 3 days after setup and continued to appear for up to 2 weeks. A 2:1 complex to well solution ratio would typically give fewer and larger crystals in the drops. Topo-97(ΔRS2) with symmetric DNA crystals started to appear within minutes of setting up the trays in 1:1 or 2:1 well to complex ratio. Subsequently it was discovered that adding phosphotungstic acid (H$_3$PW$_{12}$O$_{40}$) at between 12.5 and 15 µM made the crystals grow slower and larger and the morphology changed from plates to box-like crystals. Exact conditions for the crystallization experiments are shown in *Table 1*.

For data collection, crystals were first transferred to cryoprotectant (see *Table 1*) and then flash frozen in liquid nitrogen. All diffraction data were collected at the Life Science Collaborative Access Team station (LS CAT) at the Advanced Photon Source (APS) in Argonne National Laboratory. All data were processed using XDS (*Kabsch, 1993*), aimless (*Evans and Murshudov, 2013*), autoPROC (*Vonrhein et al., 2011*), and other programs of the CCP4 suite (*Winn et al., 2011*). Data anisotropy was analyzed with the StarAniso server (*Tickle et al., 2018*) or with autoPROC (*Vonrhein et al., 2011*).

The structure of Topo-97(ΔRS2) with an asymmetric 38 bp DNA was solved by a combination of Molecular Replacement and heavy atom phasing. A weak Molecular Replacement solution with Phaser (*McCoy et al., 2007*) was found against a 6 Å dataset using a 61 kDa fragment of topoisomerase V (*Taneja et al., 2006*). The Molecular Replacement solution did not show any additional protein regions or DNA. A phosphotungstic acid derivative was prepared by soaking crystals in 1 mM phosphotungstic acid for 2 min before flash freezing them. The Molecular Replacement model was used to locate the heavy atoms. Phasing was done with SHARP (*Vonrhein et al., 2007*; *Table 2*). The map, even at 6 Å resolution, show the position of two protein monomers and two double stranded DNA oligonucleotides. The model of a 97 kDa fragment of topoisomerase V (*Rajan et al., 2016*) was used to build most of the protein. The DNA molecules were built starting from idealized DNA. Subsequently, a higher resolution, anisotropic dataset with DNA without the abasic site was used to refine the structure to 3.24 Å resolution in the best direction using Buster (*Roversi et al., 1996*) and Phenix (*Adams et al., 2010*). Manual model building was done using Coot (*Emsley and Cowtan, 2004*; *Emsley et al., 2010*). The final model consists of two protein monomers, one full DNA molecule, and two half-length DNA molecules. The full DNA molecule, which spans 42 nucleotides per strand with 40 of them forming base pairs, binds both protein monomers while the other two DNA molecules bind one protein monomer each and sit on crystallographic axes generating crystallographic dimers made of symmetric full-length DNA molecules each with two protein monomers (*Figure 1—figure supplement 1*). The two protein complexes are not identical, as the DNA sits differently in each one. The final $R_{work}$ and $R_{free}$ for the model are 23.35% and 26.38%, respectively. The model has excellent

stereochemistry. Molprobity (*Davis et al., 2004*) shows that 97.22% of the residues in the Ramachandran plot are in favored regions and no residues are in disallowed regions. The model has an rmsd of 0.004 Å for bond lengths and 0.54° for bond angles (*Table 4*).

The structure of Topo-97(ΔRS2) with a symmetric 38 bp DNA was solved by heavy atom phasing using a 0.8 nm positively charged undecagold (Nanoprobes, Inc, Yaphank, NY) derivative (*Table 3*). The undecagold derivative was prepared by soaking crystals in 350–400 μM undecagold for 6–8 min before flash freezing them. Data from five different crystals were merged using Blend (*Foadi et al., 2013*) to create a single derivative dataset. Phenix (*Adams et al., 2010*) was used to locate a single heavy atom. Phasing was done with SHARP (*Vonrhein et al., 2007*) treating the undecagold as a spherically averaged cluster. The electron density map showed two protein monomers bound to a single 38 bp oligonucleotide. The model of Topo-97(ΔRS2) in complex with an asymmetric 38 bp DNA was used to build the two protein monomers whereas the DNA was built starting from ideal DNA. The structures of Topo-97(ΔRS2) with symmetric 39 and 40 bp symmetric DNA were obtained by refining the Topo-97(ΔRS2) in complex with an asymmetric 38 bp DNA. In all cases, anisotropic datasets were used to refine the structures using Buster (*Roversi et al., 1996*) and Phenix (*Adams et al., 2010*). Manual model building was done using Coot (*Emsley and Cowtan, 2004*; *Emsley et al., 2010*). Unlike the asymmetric complex, the symmetric complexes show the position of the abasic sites. In each case, the final model consists of two protein monomers and one full DNA molecule. Whereas the conformation of the protein is very similar in all cases, the DNA shows different distortions around the abasic sites. The final $R_{work}/R_{free}$ for the Topo-97(ΔRS2) with symmetric 38, 39, and 40 bp DNA models are 25.71%/28.77%, 21.84%/25.11%, and 22.89%/27.00%, respectively. All models have excellent stereochemistry. *Figure 5—figure supplement 1* shows density around the active site region. Molprobity (*Davis et al., 2004*) shows that 96%, 95.8%, and 97.5% of the residues in the Ramachandran plot are in favored regions and no residues are in disallowed regions for the 38, 39, and 40 bp DNA models. The models have an rmsd of 0.002 Å for bond lengths and 0.42–0.49° for bond angles (*Table 4*).

## Figures

Figures for the atomic models were created using Pymol (*DeLano, 2002*) and Chimera (*Pettersen et al., 2004*). Electrostatic surfaces were calculated using the program APBS (*Baker et al., 2001*).

## Accession numbers

Atomic coordinates and structure factors for the reported crystal structures have been deposited with the Protein Data Bank under the accession numbers 8DF7, 8DF8, 8DF9, and 8DFB.

## Acknowledgements

We thank E Smith and V Tokars for comments on the manuscript, H Mangalapalli for help with activity assays and E Campos Chavez and A Grigorescu for help with the DLS experiments as well as other members of the Mondragón laboratory for help and suggestions. Research was supported by the NIH (R35-GM118108 to AM). We acknowledge the help from the Northwestern University Structural Facility and the beamline scientists at LS-CAT/Sector 21 at the Advanced Photon Source, Argonne National Laboratory. LS-CAT/Sector 21 was supported by the Michigan Economic Development Corporation and the Michigan Technology Tri-Corridor. Support from the RH Lurie Comprehensive Cancer Center of Northwestern University to the Structural Biology Facility and the Keck Biophysics Facility is acknowledged.

## Additional information

### Funding

| Funder | Grant reference number | Author |
| --- | --- | --- |
| National Institute of General Medical Sciences | R35-GM118108 | Alfonso Mondragón |

| Funder | Grant reference number | Author |
|---|---|---|
| National Cancer Institute | P30-CA060553 | Alfonso Mondragón |

The funders had no role in study design, data collection and interpretation, or the decision to submit the work for publication.

### Author contributions

Amy Osterman, Investigation, Methodology, Writing - review and editing; Alfonso Mondragón, Conceptualization, Supervision, Funding acquisition, Investigation, Methodology, Writing - original draft, Project administration, Writing - review and editing

### Author ORCIDs

Alfonso Mondragón  http://orcid.org/0000-0002-0423-6323

### Decision letter and Author response

Decision letter https://doi.org/10.7554/eLife.72702.sa1
Author response https://doi.org/10.7554/eLife.72702.sa2

## Additional files

### Supplementary files

• Transparent reporting form

### Data availability

Atomic coordinates and structure factors for the reported crystal structures have been deposited with the Protein Data Bank under the accession numbers 8DF7, 8DF8, 8DF9, and 8DFB.

The following datasets were generated:

| Author(s) | Year | Dataset title | Dataset URL | Database and Identifier |
|---|---|---|---|---|
| Mondragón A, Osterman A | 2022 | Structures of topoisomerase V in complex with DNA reveal unusual DNA binding mode and novel relaxation mechanism | https://www.rcsb.org/structure/8DF7 | RCSB Protein Data Bank, 8DF7 |
| Mondragón A, Osterman A | 2022 | Structures of topoisomerase V in complex with DNA reveal unusual DNA binding mode and novel relaxation mechanism | https://www.rcsb.org/structure/8DF8 | RCSB Protein Data Bank, 8DF8 |
| Mondragón A, Osterman A | 2022 | Structures of topoisomerase V in complex with DNA reveal unusual DNA binding mode and novel relaxation mechanism | https://www.rcsb.org/structure/8DF9 | RCSB Protein Data Bank, 8DF9 |
| Mondragón A, Osterman A | 2022 | Structures of topoisomerase V in complex with DNA reveal unusual DNA binding mode and novel relaxation mechanism | https://www.rcsb.org/structure/8DFB | RCSB Protein Data Bank, 8DFB |

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
