## [Editor Report]

This is a valuable paper with convincing data. The work presents the first structure of Methanopyrus kandleri Topoisomerase V bound to DNA, revealing two important features of the enzyme's mechanism. The first is that the active site toggles between opened, DNA-accessible state to a closed state where the active site cleft is blocked and inaccessible to nucleic acid. The second is a striking array of helix-hairpin-helix motifs that wrap the duplex DNA. The findings will be of interest to researchers working on understanding structure/function relationships of nucleic acid enzymes, particularly in the topoisomerase and DNA repair fields.

---

## [Decision Letter]

**Decision letter after peer review:**

Thank you for submitting your article "Structures of topoisomerase V in complex with DNA reveal unusual DNA binding mode and novel relaxation mechanism" for consideration by *eLife*. Your article has been reviewed by 2 peer reviewers, and the evaluation has been overseen by a Reviewing Editor and Cynthia Wolberger as the Senior Editor. The following individual involved in review of your submission has agreed to reveal their identity: Anthony Maxwell (Reviewer #3).

Essential revisions:

Both reviewers agree that this work represents a significant milestone in characterizing type IC topoisomerases but also feel that findings need additional corroborating studies. These include:

(1) Investigation of mutants predicted to affect DNA bending and opening of the active-site cleft.

(2) Corroboration of DNA bending and proposed protein conformational changes by SAXS or FRET techniques. Do mutations in the hinge region of the protein alter topoisomerase activity?

(3) Some discussion of the two DNA segments bound to the HLH region that are present in the asymmetric structure. These seem reminiscent of DNA a crossover. Does mutagenesis of these interaction sites impact enzyme function?

(4) The symmetric form shows what looks to be a dimer, with an intertwining of the HLH elements that appears very complementary. E.g., there are interesting arrays salt bridging interactions between the dimers. In addition, the HhH from one monomer can be seen to contact the catalytic domain of the opposite dimer with interesting metal (K^+^) bridges. Overall, the dimer should be detailed more in the manuscript, as well as investigated biochemically for its potential relevance.

(5) Are the observed HhH DNA interaction interfaces required for AP lyase activity? This should be assessed by mutating the surfaces and measuring this enzymatic function.

While not every request need be addressed in a revision, a majority should be. It is recognized that the additional requested studies are sufficiently extensive that you may wish to submit the work elsewhere.

*Reviewer #1 (Recommendations for the authors):*

1. Aspects of the protein conformational change and DNA binding surfaces should be probed by mutagenesis and any of a number of biochemical approaches to evaluate their roles in topoisomerase conformational dynamics and activity, and AP lyase activity.

2. The discussion of the conformational changes would benefit from structural overlays of the intact protein. The illustrations in Figure2 -supplement 2 are informative, but the details are missing. The overall conformational change is also unclear from the diagrams. What interactions stabilize the opened versus closed conformations. Can the significance and mechanism of this transition be evaluated by a mutational strategy? Likewise better diagrams of the protein-DNA interaction surfaces could better support the authors discussion.

3. The paper is reasonably well written. The discussion on page 4 of the crystallization strategy and results could be condensed.

*Reviewer #2 (Recommendations for the authors):*

Although this work is an important milestone in topoisomerase structure-function, it does not give as much insight as might have been hoped for; there are still significant uncertainties. As far as I can see the structural work has been competently done and is comprehensively reported.

---

## [Author Response]

Essential revisions:Both reviewers agree that this work represents a significant milestone in characterizing type IC topoisomerases but also feel that findings need additional corroborating studies. These include:(1) Investigation of mutants predicted to affect DNA bending and opening of the active-site cleft.

We made 19 mutants in three different areas: in the linker helix, in the positively charged groove where the DNA bends, and near the exit point of the DNA. Some results stand out: (i) one mutant, L290P, which was designed to prevent helix formation around the break point, almost completely abolished DNA relaxation activity, (ii) an arginine, R108, wedged between the cleaved and non-cleaved DNA strands and near the active site, is essential for activity. It is likely that this arginine allows the two highly bent strands to come closer together, and (iii) single mutants change the positively charged residues in the DNA binding groove have a modest effect, but double mutants led to a significant loss of DNA relaxation activity. This suggest that, as the structure suggests, the charge character of this region is important to accommodate the bent DNA. The mutagenesis results confirm that the exposure of the active site through changes in the linker region is essential, that the charge character of the groove by the active site is needed to bend the DNA, and, not surprising, that interactions with the non-cleaved strand are also important for activity. Overall, the mutagenesis results confirm and validate the structural findings and support the conclusions.

The mutagenesis results are in a new section in the Results section and are part of the Discussion section. Two new figures, new Figure 4 and new Figure 4 —figure supplement 1 summarize the results and show typical data.

(2) Corroboration of DNA bending and proposed protein conformational changes by SAXS or FRET techniques. Do mutations in the hinge region of the protein alter topoisomerase activity?

As mentioned above, we investigated the importance of the linker helix by mutagenesis together with DNA relaxation assays. While mutating some residues facing the DNA had some effect on DNA relaxation activity, the most dramatic effect was observed with a mutant in the hinge region designed to disrupt helix formation. This mutant, L290P, had almost no activity and confirms the importance of the conformational change of this helix in the mechanism.

Corroborating DNA bending by FRET is not possible in our complex due to the distance between the two ends of the bent DNA molecule, an obvious place to attach dyes without disrupting the structure. The distance between the ends of the DNA is around 110Å, which is much longer than the Forster R_0_ values for most dye pairs, normally in the 30 – 60 Å range. This means that a FRET experiment would be insensitive to the changes expected. Although we would like to measure the bending extent in solution, this is not easily done and would require a major and new experimental approach.

(3) Some discussion of the two DNA segments bound to the HLH region that are present in the asymmetric structure. These seem reminiscent of DNA a crossover. Does mutagenesis of these interaction sites impact enzyme function?

We agree that the presence of a second DNA segment is reminiscent of a crossover point. We had not discussed it before as we deemed it too speculative. We are glad to see that the reviewers agree that this is an important point to discuss. We have added a new paragraph in the Discussion section to address this point and its implications. A revised Figure 6 now includes a cartoon to illustrate how two bound DNA segments would resemble a crossover point.

(4) The symmetric form shows what looks to be a dimer, with an intertwining of the HLH elements that appears very complementary. E.g., there are interesting arrays salt bridging interactions between the dimers. In addition, the HhH from one monomer can be seen to contact the catalytic domain of the opposite dimer with interesting metal (K^+^) bridges. Overall, the dimer should be detailed more in the manuscript, as well as investigated biochemically for its potential relevance.

We agree that the dimers in the crystal are very intriguing, but we feel that they are not biologically relevant for a major reason: the structure corresponds to a 97 kDa fragment that is missing the last two (HhH)_2_ domains. Inspection of the dimers reveals that the packing would preclude binding of the additional two (HhH)_2_ domains. To illustrate this, we modeled the last two domains and showed that their presence could not be accommodated. In addition, we measured the oligomerization state of the protein/DNA complex in solution. Unfortunately the complex of the full length protein or the 97 kDa fragment with DNA falls apart in a gel filtration column, precluding measuring the molecular weight in solution by SEC-MALS. For this reason, we used Dynamic Light Scattering (DLS) after carefully optimizing the conditions to make the complex (discussed in the Materials and methods section). We could not measure the properties of the full length protein as it has residual aggregates even at high salt concentration, but we succeeded in measuring the 97 kDa fragment solution properties by DLS. The DLS experiments suggest that in solution the 97 kDa fragment most likely forms a 1:1 complex with DNA. The DLS measurements of the free protein agree remarkably well with previous measurements from SAXS, supporting the use of DLS for the experiments. These findings are discussed in the Results section and a new figure showing the modeling of the full length protein as a dimer and the clashes present is in new Figure 2 —figure supplement 4.

(5) Are the observed HhH DNA interaction interfaces required for AP lyase activity? This should be assessed by mutating the surfaces and measuring this enzymatic function.

In the past we have characterized the AP lyase activity and one of our findings was that topoisomerase V retains AP lyase activity even when several residues in the vicinity of each of the AP lyase active sites are mutated. It takes several mutations to remove activity as topoisomerase V seems very forgiving to changes around the AP lyase sites. For example, the fragment that we used has 6 residues mutated to alanine to ensure that there is not activity left in that (HhH)_2_ domain. We do not see interactions between the DNA abasic sites and the (HhH)_2_ domains known to be implicated in AP lyase activity, making it difficult to pinpoint good residues to mutate. While we agree that investigating more the implications for AP lyase activity is important, this would be a major effort and outside the scope of the manuscript.

Reviewer #1 (Recommendations for the authors):1. Aspects of the protein conformational change and DNA binding surfaces should be probed by mutagenesis and any of a number of biochemical approaches to evaluate their roles in topoisomerase conformational dynamics and activity, and AP lyase activity.

We probed the role of the linker helix and the DNA binding surfaces on DNA relaxation activity using site directed mutagenesis. Overall, the results are in agreement with the observations from the structure. We did not study the AP lyase activity as we feel this is outside the scope of this manuscript.

2. The discussion of the conformational changes would benefit from structural overlays of the intact protein. The illustrations in Figure2 -supplement 2 are informative, but the details are missing. The overall conformational change is also unclear from the diagrams. What interactions stabilize the opened versus closed conformations. Can the significance and mechanism of this transition be evaluated by a mutational strategy? Likewise better diagrams of the protein-DNA interaction surfaces could better support the authors discussion.

We revised some of the figures for clarity, in particular Figure 2 —figure supplement 2. We also added a new Figure, Figure 2 —figure supplement 3 that shows an overlay of the structures to emphasize the conformational changes and also highlights the interface and the extent of the buried surface area in the open and closed conformations. It serves to illustrate the conformational changes better. A description of the nature of the domain/domain interfaces has been included in the text.

3. The paper is reasonably well written. The discussion on page 4 of the crystallization strategy and results could be condensed.

We have condensed significantly the crystallization section and also consolidated Table I, which describes the crystallization conditions.